# Atomic HINs: Entity-Attribute Duality for Heterogeneous Graph Modeling

**Shao-En Lin**[1], **Ming-Yi Hong**[123], **Miao-Chen Chiang**[123], **Chih-Yu Wang**[23], **Che Lin**[13*]
[1]National Taiwan University, [2]Academia Sinica,
[3]Data Science Degree Program, National Taiwan University and Academia Sinica

## Abstract

Heterogeneous Information Networks (HINs) provide a powerful framework for modeling multi-typed entities and relations, typically defined under a fixed schema. Yet, most research assumes this structure is given, overlooking the fact that alternative designs can emphasize different aspects of the data and substantially influence downstream performance. As a theoretical foundation for such designs, we introduce the principle of *entity-attribute duality*: attributes can be atomized as entities with their associated relations, while entities can, in turn, serve as attributes of others. This principle motivates *atomic HIN*, a canonical representation that makes all modeling choices explicit and achieves maximal expressiveness. Building on this foundation, we propose a systematic framework for task-specific schema refinement. Within this framework, we demonstrate that widely used benchmarks correspond to heuristic refinements of the *atomic HIN*—often far from optimal. Across eight datasets, refinement alone enables a simplified Relational GCN (sRGCN) to achieve state-of-the-art performance on node- and link-level tasks, with further gains from advanced HGNNs. These results highlight schema design as a key dimension in heterogeneous graph modeling. By releasing the *atomic HINs*, searched schemas, and refinement framework, we enable principled benchmarking and open the way for future work on schema-aware learning, automated structure discovery, and next-generation HGNNs. Our code is available at: `https://github.com/ntuidssplab/AtomHIN`.

## 1 Introduction

Heterogeneous Information Networks (HINs) provide a powerful abstraction for modeling systems with multiple types of entities and relations. Such graphs naturally arise in bibliometrics, e-commerce, knowledge graphs, and social networks, where diverse node and edge types yield rich semantics. To leverage these structures, Heterogeneous Graph Neural Networks (HGNNs) extend Graph Neural Networks (GNNs) with type-aware message passing across heterogeneous schemas.

Despite this progress, most research relies on a few benchmark HINs with manually specified schemas, often chosen heuristically. In practice, multiple valid schemas can be derived from the same data. For instance, the IMDb benchmark is constructed from a single movie table[1]: columns such as *actor* and *director* are processed to be entities, while others (e.g., keyword, language, country) remain attributes (Figure 1b). In some variants, *keywords* are treated as entities rather than mere attributes (Figure 1c), while additional unexplored variants are equally possible (Figure 1d).

The ambiguity in schema design has received limited attention. Some recent efforts (Fey et al., 2024) explore schema construction from relational databases, but they still rely on database-specific design choices, each of which corresponds to different schema variants. Consequently, the broader problem of designing HIN schemas remains open.

We introduce *atomic HIN*, grounded in the principle of *entity-attribute duality*: attributes can be atomized as entities, and entities can in turn serve as attributes of others. This duality maximizes expressiveness by making all schema choices explicit, but also increases modeling complexity.

---

*Corresponding author: chelin@ntu.edu.tw.
[1]`https://www.kaggle.com/datasets/karrrimba/movie-metadatacsv`

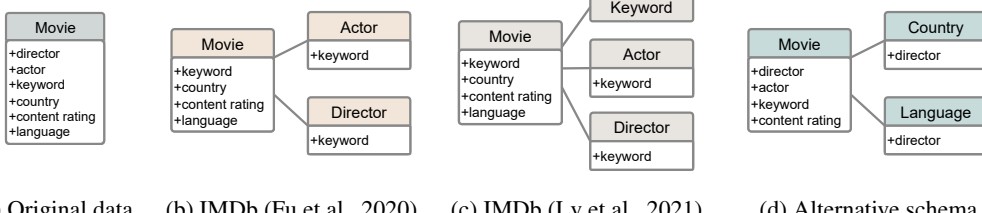

| (a) Original data | (b) IMDb (Fu et al., 2020) | (c) IMDb (Lv et al., 2021) | (d) Alternative schema |

Figure 1: Illustration of multiple heterogeneous schemas derived from the same IMDb source table.

To address this, we propose *task-specific schema refinement*, which selects or discards node types (entities) and edge types (relations) to tailor the schema. Node-type selection determines which entities are assigned unique identities, while edge-type selection removes relations of limited utility, simplifying both the schema and the model. From this perspective, widely used benchmark schemas are merely heuristic node- and edge-type selections, as essentially all schema variants can be viewed as refinements of the atomic form.

Empirically, pairing the atomic form with sRGCN—a simplified RGCN with stronger parameter sharing—achieves state-of-the-art results on eight datasets spanning node classification and link prediction. On node classification, Macro-F1 improves by up to 6.2% over recent advanced HGNNs, and on link prediction, ROC-AUC increases by an average of 4.9%. Our systematic search further shows that benchmark schemas often diverge from optimal ones, highlighting the effectiveness of *atomic HINs* with schema refinement. Moreover, schemas refined with sRGCN generalize well to advanced HGNNs, with subsequent refinement under stronger models yielding additional gains.

We summarize our contributions as follows:

- We introduce the principle of *entity–attribute duality*, which motivates the *atomic HIN*, a canonical representation that makes all schema choices explicit and achieves maximal expressiveness.

- Building on this foundation, we propose *schema refinement*, which turns the otherwise open-ended challenge of schema design into a principled structural learning problem, enabling task-aware optimization while controlling complexity.

- We show that widely used benchmark schemas are, in fact, particular refinements of the atomic form, and our systematic search reveals that these heuristic choices often differ substantially from optimal ones, underscoring the central role of schema design in HGNN development and evaluation.

- Extensive experiments on eight benchmarks demonstrate that *atomic HINs* with refined schemas consistently improve HGNN performance. Even a simplified RGCN (sRGCN) achieves state-of-the-art results, while advanced HGNNs obtain further improvements. Analysis of the search results further confirms the importance of *atomic HINs* and schema refinement.

## 2 RELATED WORK

### 2.1 HETEROGENEOUS GRAPH NEURAL NETWORKS

Heterogeneous graph neural networks (HGNNs) extend GNNs to exploit the semantic richness of heterogeneous information networks (HINs). Early models such as HAN (Wang et al., 2019) and MAGNN (Fu et al., 2020) rely on hand-crafted metapaths to define composite relations. RGCN (Schlichtkrull et al., 2018), HGT (Hu et al., 2020b), and SimpleHGN (Lv et al., 2021) instead model heterogeneity through relation-specific transformations or heterogeneous attention. GTN (Yun et al., 2019), MHGCN (Yu et al., 2022), and RE-GNN (Wang et al., 2023) further learn soft weights over edge types, effectively performing differentiable metapath or subgraph selection. More recently, SeHGNN (Yang et al., 2023), PSHGCN (He et al., 2024), and LMSPS (Li et al., 2024) precompute propagation along metapaths to support mini-batch training on large-scale HINs.

Collectively, these methods capture heterogeneous structure at the level of relations, metapaths, or soft subgraphs, using either manual design or differentiable selection. However, they all assume a *fixed* underlying schema on which message passing is defined.

## 2.2 HETEROGENEOUS GRAPH SCHEMA DESIGN

Compared to architectural innovations, the design of heterogeneous graph schemas has received relatively little systematic study. Common strategies include augmenting schemas with metapath-based edges (Wang et al., 2019; Fu et al., 2020; Hu et al., 2024), introducing metapath-derived features (Lv et al., 2021; Fey et al., 2024), and adding attribute-derived relations, typically limited to binary or categorical attributes encoded as edges.

While flexible, these strategies produce schema-design spaces that are both unbounded and complex. Attribute-derived relations expand the relation set, while metapath-based constructions grow exponentially with the number of relations. Benchmark datasets, therefore, implement these ideas inconsistently and often rely on ad-hoc decisions. For instance, HGB (Lv et al., 2021) selectively elevates certain attributes (e.g., actors, keywords) to entities while leaving others (e.g., language, country) as plain features. OGB (Hu et al., 2020a) averages textual-term embeddings into paper nodes but does not construct term nodes or relations. RelBench (Fey et al., 2024) moves toward systematic schema construction while still yielding multiple valid choices.

In contrast, we introduce a **simple and principled framework for schema design**. The atomic HIN unifies and generalizes existing ad-hoc practices into a canonical representation in which *all* schema decisions are explicit. On this foundation, schema refinement becomes a systematic and optimizable process: it reduces the originally open-ended design space to tractable selections of nodes and relation types without sacrificing the expressive power available to downstream HGNNs. This positions schema design not merely as a preprocessing step but as a core dimension of heterogeneous graph modeling, enabling principled benchmarking and schema-aware model development.

## 3 PRELIMINARIES

**Attributed Heterogeneous Information Networks**

An undirected HIN is a graph $\mathcal{G} = (\mathcal{V}, \mathcal{E})$ consisting of a node set $\mathcal{V}$ and an edge set $\mathcal{E}$. Each node $v \in \mathcal{V}$ and edge $e \in \mathcal{E}$ is assigned a type through mappings $\phi : \mathcal{V} \to \mathcal{T}$ and $\psi : \mathcal{E} \to \mathcal{R}$, where $\mathcal{T} = \{\tau\}_{\tau=1}^{|\mathcal{T}|}$ and $\mathcal{R} = \{r\}_{r=1}^{|\mathcal{R}|}$ denote the sets of node types (entities) and edge types (relations). Edges are represented by adjacency matrices $\{\boldsymbol{A}_r \in \mathbb{R}^{|\mathcal{V}| \times |\mathcal{V}|}\}_{r=1}^{|\mathcal{R}|}$, where $\boldsymbol{A}_r$ corresponds to edges with relation $r$ with $\{e \in \mathcal{E} \mid \psi(e) = r\}$.

An attributed HIN is further associated with an attribute set $\mathcal{F}$, where each $f \in \mathcal{F}$ is assigned to an owner node type via $\zeta : \mathcal{F} \to \mathcal{T}$. Attributes are then represented as feature matrices $\{\boldsymbol{X}_f \in \mathbb{R}^{|\mathcal{V}_{\zeta(f)}| \times d_f}\}_f$, where $\mathcal{V}_{\zeta(f)} = \{v \in \mathcal{V} \mid \phi(v) = \zeta(f)\}$ and $d_f$ is the dimension of $f$.

**Spectral heterogeneous graph convolution (SHGC).** Defferrard et al. (2016) showed that spectral filters on homogeneous graphs can be parameterized as $L$-order polynomials of the Laplacian eigenvalues. Butler et al. (2023) extended the formulation to heterogeneous graphs, introducing spectral filters based on non-commutative polynomials over relation-specific shift operators. For an input signal $\boldsymbol{x} \in \mathbb{R}^{|\mathcal{V}|}$, the filter is defined as

$$\boldsymbol{H}(\boldsymbol{S}_1, \ldots, \boldsymbol{S}_{|\mathcal{R}|}; \Theta)\boldsymbol{x} = \theta_0 \boldsymbol{I} \boldsymbol{x} + \sum_{\ell=1}^{L} \sum_{r_1, \ldots, r_\ell} \theta_{r_1, \ldots, r_\ell} (\boldsymbol{S}_{r_1} \cdots \boldsymbol{S}_{r_\ell}) \boldsymbol{x}, \tag{1}$$

where $\boldsymbol{S}_r$ is the shift operator for relation $r$. A common choice is the row-normalized adjacency $\tilde{\boldsymbol{A}}_r = \boldsymbol{D}_r^{-1} \boldsymbol{A}_r$, where $D_r[i,i] = \sum_{j=1}^{|\mathcal{V}|} |A_r[i,j]|$ is the degree matrix of $\boldsymbol{A}_r$. The learnable parameters are collected in $\Theta = \{\theta_0\} \cup \{\theta_{r_1, \ldots, r_\ell}\}_{r_1, \ldots, r_\ell}$, where each coefficient $\theta_0, \theta_{r_1, \ldots, r_\ell} \in \mathbb{R}$. In this paper, we adopt SHGC as a general formulation of HGNNs.

## 4 METHODOLOGY

### 4.1 FROM ATTRIBUTE ATOMIZATION TO ATOMIC HINS

Constructing graph structure from attributes is a long-standing but often implicit technique in HIN schema design. In most existing pipelines, this is applied during preprocessing only to binary or

categorical attributes, which can be expanded into one-hot or multi-hot representations and then converted into additional node and edge types. A widely used example is the IMDb dataset (Figure 1), where attributes such as keywords or actors are elevated to entities.

Formally, this process operates independently on each attribute. In this work, we generalize the idea beyond binary attributes, treating all attributes uniformly under the principle of entity–attribute duality. For a given attribute, we define it as follows:

**Definition 4.1** (Attribute Atomization). *Given an attribute $f \in \mathcal{F}$ of a HIN $\mathcal{G}$ with feature matrix $\boldsymbol{X}_f$, atomizing $f$ produces an augmented HIN $\mathcal{G}' = (\mathcal{V} \cup \mathcal{U}, \mathcal{E} \cup \mathcal{E}_f)$, where $\mathcal{U} = \{u_1, \ldots, u_{d_f}\}$ is the set of new nodes with cardinality corresponding to attribute dimensions and*

$$\mathcal{E}_f = \big\{(v_i, u_j) \mid X_f[i,j] \neq 0,\ 1 \le i \le |\mathcal{V}_\tau|,\ 1 \le j \le d_f \big\}.$$

*Here $v_i$ denotes the $i$-th node of $\mathcal{V}_{\zeta(f)}$ and $u_j$ is the $j$-th node of the induced set $\mathcal{U}$. Each edge $(v_i, u_j)$ is weighted by $X_f[i,j]$. This introduces a new node type $\tau'$ and a new edge type $r'$, yielding $\mathcal{T}' = \mathcal{T} \cup \{\tau'\}$ with $\phi'(v) = \tau'$ if $v \in \mathcal{U}$ and $\phi'(v) = \phi(v)$ otherwise; and $\mathcal{R}' = \mathcal{R} \cup \{r'\}$ with $\psi'(e) = r'$ if $e \in \mathcal{E}_f$ and $\psi'(e) = \psi(e)$ otherwise.*

In this way, atomization replaces the nonzero entries of $\boldsymbol{X}_f$ with explicit edges to the corresponding attribute-induced nodes, thereby converting attributes—binary, categorical, or numerical—into explicit structure and relations. This allows HGNNs to exploit structural dependencies and relational patterns such as metapaths (Sun et al., 2011).

The common practice of constructing edges via ID matching across relational tables (Fey et al., 2024) is a special case where $\boldsymbol{X}_f$ is sparse and binary, producing a small number of attribute nodes. When $\boldsymbol{X}_f$ encodes numerical attributes, atomization produces the same star-like pattern: each attribute dimension becomes an induced node, and every original node connects to it with a single weighted edge. Thus, both binary and numerical attributes yield structurally similar, star-shaped adjacency patterns centered at attribute-induced nodes.

Applying atomization to all attributes yields the *atomic HIN*, a canonical representation in which all information is expressed structurally, achieving maximal expressiveness and making explicit all modeling choices. This representation serves as the foundation for our next step: task-specific schema refinement.

## 4.2 Schema Refinement via Node- and Edge-Type Selection

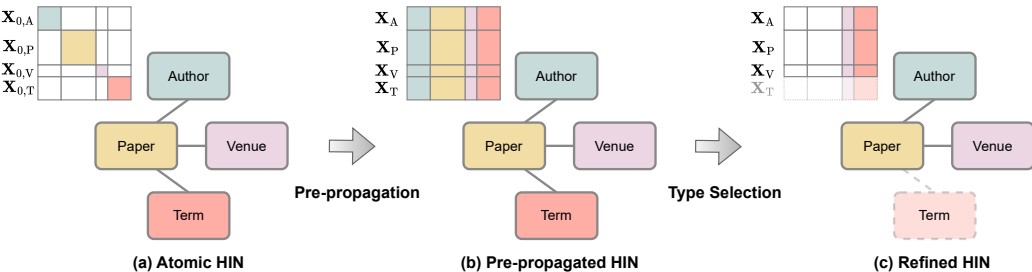

Figure 2: Toy example of schema refinement on *atomic HINs*: pre-propagation ensures independence between node-type and edge-type selection.

We then propose the schema refinement problem, a binary selection problem over node types (entities) and edge types (relations), aiming to harness the expressiveness of *atomic HINs* while controlling complexity. To this end, we first define two basic operations:

- **Node-type selection.** For each node type $\tau \in \{1, \ldots, |\mathcal{T}|\}$, a binary parameter $\beta_\tau \in \{0, 1\}$ determines whether nodes of this type are assigned a unique identity and thus contribute as attributes to the HIN.

- **Edge-type selection.** For each relation $r \in \{1, \ldots, |\mathcal{R}|\}$, a binary parameter $\alpha_r \in \{0, 1\}$ specifies whether the relation is retained and modeled in the schema.

Under the SHGC framework (Equation 1), which we use as a general form of HGNNs, refinement on an *atomic HIN* is expressed as:

$$\bar{\boldsymbol{Z}} = \boldsymbol{H}\big(\alpha_1 \boldsymbol{S}_1, \ldots, \alpha_{|\mathcal{R}|} \boldsymbol{S}_{|\mathcal{R}|}\big) \boldsymbol{X}(\beta_1, \ldots, \beta_{|\mathcal{T}|}), \tag{2}$$

where $\boldsymbol{H}(\cdot)$ denotes the SHGC filter, and $\boldsymbol{X}(\beta_1, \ldots, \beta_{|\mathcal{T}|}) \in \mathbb{R}^{|\mathcal{V}| \times |\mathcal{V}|}$ is the feature matrix after node-type selection.

The edge-type selectors $\alpha_r$ directly control which relations and edges participate in message passing. Setting $\alpha_r = 0$ removes relation $r$ and its edges entirely, equivalent in practice to dropping that relation from the constructed HIN. This makes refinement compatible with any HGNN in a plug-and-play manner, without requiring architectural modifications.

Node-type selection specifies which node types are assigned unique learnable embeddings. A straightforward construction is to assign the identity matrix as node features for the selected node types:

$$\boldsymbol{X}_0(\beta_1, \ldots, \beta_{|\mathcal{T}|}) = \beta_1 \hat{\boldsymbol{I}}_1 + \beta_2 \hat{\boldsymbol{I}}_2 + \cdots + \beta_{|\mathcal{T}|} \hat{\boldsymbol{I}}_{|\mathcal{T}|}, \tag{3}$$

where $\hat{\boldsymbol{I}}_\tau \in \mathbb{R}^{|\mathcal{V}| \times |\mathcal{V}|}$ is the type-specific identity matrix, with $\hat{I}_\tau[i, i] = 1$ if the $i$-th node $v_i \in \mathcal{V}$ has type $\phi(v_i) = \tau$, and 0 otherwise. When all types are selected, $\boldsymbol{X}_0(1, \ldots, 1) = \boldsymbol{I}_{|\mathcal{V}|}$. The motivation is to learn embeddings only for informative node types, thereby reducing parameters and mitigating overfitting.

However, this naive construction can induce *dependencies*.

**Definition 4.2** (Dependency relative to a node type). *For a node type $\tau_i \in \{1, \ldots, |\mathcal{T}|\}$, we say that $\tau_i$ has a* dependency *on $\tau_j$ if $\hat{\boldsymbol{I}}_{\tau_i} \bar{\boldsymbol{Z}}_0 \hat{\boldsymbol{I}}_{\tau_j} = \boldsymbol{0}$ for some $\tau_j$ with $\beta_{\tau_j} = 1$, where $\bar{\boldsymbol{Z}}_0 = \boldsymbol{H}(\cdot) \boldsymbol{X}_0(\cdot)$ is the output of SHGC on the naive feature initialization.*

Dependencies arise because pruning certain edge types can disconnect selected nodes from the rest of the graph. For example, in Figure 2, if all edges incident to *term* nodes are removed, their embeddings become isolated and cannot contribute to downstream predictions. Thus, although edge-type pruning improves efficiency and compatibility, it may inadvertently invalidate node-type selections by cutting off identity propagation.

To mitigate this issue, we introduce a pre-propagation feature initialization, which ensures that embeddings of selected node types remain accessible regardless of subsequent edge-type choices.

## 4.3 PRE-PROPAGATION FEATURE INITIALIZATION

To address the dependency issue in Definition 4.2, we introduce a pre-propagation feature initialization. The intuition is straightforward: prior to refinement, each selected node type propagates its identity once to other node types, ensuring that its signal remains accessible even if incident edge types are subsequently removed. Formally, we define pre-propagated features as

$$\boldsymbol{X}(\beta_1, \ldots, \beta_{|\mathcal{T}|}) = \Big(\boldsymbol{I} + \sum_{\tau_i \neq \tau_j} \tilde{\boldsymbol{A}}_{\langle \tau_i, *, \tau_j \rangle}\Big) \boldsymbol{X}_0(\beta_1, \ldots, \beta_{|\mathcal{T}|}), \tag{4}$$

where $\tilde{\boldsymbol{A}}_{\langle \tau_i, *, \tau_j \rangle}$ denotes the adjacency of the shortest path from type $\tau_j$ to $\tau_i$, defined as the shortest product of adjacency matrices satisfying $\hat{\boldsymbol{I}}_{\tau_i} \tilde{\boldsymbol{A}}_{\langle \tau_i, *, \tau_j \rangle} \hat{\boldsymbol{I}}_{\tau_j} \neq 0$. We provide details for the implementation of this pre-propagation step in Appendix B.4.

This initialization effectively distributes each identity embedding beyond its original source node, ensuring its availability across node types and thereby eliminating hidden dependencies.

**Lemma 4.1** (Independence of Selections). *With pre-propagation, for any node type $\tau_i$ and any selected type $\tau_j$ with $\beta_{\tau_j} = 1$, $\hat{\boldsymbol{I}}_{\tau_i} \bar{\boldsymbol{Z}} \hat{\boldsymbol{I}}_{\tau_j} \neq \boldsymbol{0}$. Hence, node-type selection is independent of edge-type selection: even if all edges incident to $\tau_j$ are removed, no dependency arises.*

**Lemma 4.2** (Neutrality of Pre-propagation). *Consider an SHGC with row-normalized adjacencies $\boldsymbol{S}_r = \tilde{\boldsymbol{A}}_r$ as shift operators and convolution order $L$ sufficiently large. If raw identity features $\boldsymbol{X}_0(1, \ldots, 1) = \boldsymbol{I}$ are replaced with pre-propagated features $\boldsymbol{X}(1, \ldots, 1)$, the effect is merely a reparameterization of the filter coefficients $\{\theta_{r_1, \ldots, r_\ell}\}_{r_1, \ldots, r_\ell}$. Hence, pre-propagation does not alter the expressive power of the model.*

Proofs are deferred to Appendix B.

In summary, pre-propagation resolves node dependencies (Lemma 4.1) without reducing expressive power (Lemma 4.2), ensuring that edge types can be pruned without invalidating node type selections. We illustrate this with row-normalized adjacency, while in practice we adopt a generalized union via element-wise maximum (Lv et al., 2021) (see Appendix B.3). Pre-propagation thus provides a principled basis for schema refinement without manual adjustments for each choice.

## 4.4 SYSTEMATIC SEARCH FOR SCHEMA REFINEMENT

Schema refinement reduces to a binary selection over node and edge types, but the search space of $2^{|\mathcal{R}|+|\mathcal{T}|}$ candidates is prohibitively large. Moreover, the space is highly skewed: retaining more edge types generally increases expressiveness (Lemma 4.3), while sparse or high-cardinality node types often introduce excessive parameters and risk overfitting. Hence, naive grid or random search is ineffective.

To address this, we formulate schema refinement as a hyperparameter optimization problem and adopt a genetic algorithm (GA)-based search strategy. While GAs have been widely studied for binary optimization, to our knowledge, they have not been applied to HIN schema refinement. Our formulation naturally enables this approach, providing a practical and effective means to explore the skewed binary space and achieve near-optimal solutions within reasonable budgets (Deb et al., 2002; Katoch et al., 2021). We initialize the population with the vanilla schema and jointly optimize schema parameters with model depth $L$, consistent with Lemma 4.2. Full algorithmic details are provided in Appendix A.4.

## 4.5 HGNNS FOR ATOMIC HINS

Atomizing attributes enhances expressiveness by establishing additional structure and relations, enabling HGNNs to capture richer semantics. In particular, the new relations introduced through attribute atomization enlarge the filter space, as formalized below:

**Lemma 4.3** (Attribute Atomization Enlarges Filter Space). *Let $\mathcal{R}$ be the original set of edge types, and let $\mathcal{R}' \supset \mathcal{R}$ denote the set obtained after attribute atomization. Under the SHGC formulation, the space of heterogeneous filters spanned by*

$$\{\boldsymbol{I}\} \cup \{\boldsymbol{S}_{r_1} \cdots \boldsymbol{S}_{r_\ell} \mid r_1, \ldots, r_\ell \in \mathcal{R}', 1 \leq \ell \leq L\}$$

*is strictly larger than the corresponding space defined with $\mathcal{R}$. Hence, attribute atomization strictly enlarges the filter space by converting attributes into entities with associated relations.*

To connect with existing HGNNs and illustrate the broad utility of *atomic HINs*, we observe that many architectures can be viewed as instances or approximations of SHGC.

**Proposition 4.1** (RGCN (Schlichtkrull et al., 2018) as a First-Order Approximation of SHGC). *RGCN can be expressed as a first-order approximation of SHGC under row-normalized adjacencies $\boldsymbol{S}_r = \tilde{\boldsymbol{A}}_r$, with filter coefficient matrices $\{\boldsymbol{W}_0\} \cup \{\boldsymbol{W}_r^{(\ell)}\}_{r,\ell}$.*

**Proposition 4.2** (GTN (Yun et al., 2019) as a First-Order Approximation of SHGC). *GTN can be expressed as a first-order approximation of SHGC under row-normalized adjacencies, with scalar filter coefficients $\{\theta_0\} \cup \{\theta_r^{(\ell)}\}_{r,\ell}$.*

These cases show that widely used HGNNs, despite differing in interpretation, are first-order SHGC with distinct parameter-sharing schemes (e.g., GTN enforces $\boldsymbol{W}_r^{(\ell)} = \theta_r^{(\ell)}\boldsymbol{I}$). Along with additional representative models such as SimpleHGN (Lv et al., 2021), SeHGNN (Yang et al., 2023), and PSHGCN (He et al., 2024), detailed derivations and extensions to higher-order variants are provided in Appendix C.

On *atomic HINs*, SHGC particularly favors architectures with stronger parameter sharing (e.g., GTN-style), since all inputs reduce to unique identity embeddings. In this setting, heavy feature transformations or MLP layers become largely redundant—consistent with the empirical findings of He et al. (2020) that shallow parameterization can outperform deeper transformations when natural features are absent.

Table 1: Performance comparison on different datasets. Top panel reports **node classification** results; bottom panel reports **link prediction** results. Best results are in **bold**, second-best are underlined. Statistical significance is marked by $^\dagger$ ($p < 0.001$) and $^\ddagger$ ($p < 0.01$). OOM indicates out-of-memory.

| HGNN | IMDb | | Freebase | | DBLP | | ACM | | OGBN-MAG | |
|---|---|---|---|---|---|---|---|---|---|---|
| | Macro-F1 | Micro-F1 | Macro-F1 | Micro-F1 | Macro-F1 | Micro-F1 | Macro-F1 | Micro-F1 | Acc. (Val) | Acc. (Test) |
| XGBoost | 64.27±0.21 | 67.58±0.20 | – | – | 77.96±0.37 | 78.65±0.34 | 86.92±0.32 | 86.90±0.30 | 21.85±0.16 | 22.65±0.22 |
| RGCN | 58.85±0.26 | 62.05±0.15 | 46.78±0.77 | 58.33±1.57 | 91.52±0.50 | 92.07±0.50 | 91.55±0.74 | 91.41±0.75 | 48.35±0.36 | 47.37±0.48 |
| GTN | 66.35±0.26 | 68.93±0.34 | 46.60±2.25 | 63.72±1.01 | 93.97±0.18 | 94.43±0.17 | 93.32±0.17 | 93.24±0.18 | OOM | OOM |
| HGT | 63.00±1.19 | 67.20±0.57 | 29.28±2.52 | 60.51±1.16 | 93.01±0.23 | 93.49±0.25 | 91.12±0.76 | 91.00±0.76 | 49.89±0.47 | 49.27±0.61 |
| SimpleHGN | 63.53±1.36 | 67.36±0.57 | 47.72±1.48 | 66.29±0.45 | 94.01±0.24 | 94.46±0.22 | 93.42±0.44 | 93.35±0.45 | OOM | OOM |
| HINormer | 64.65±0.53 | 67.83±0.34 | 52.18±0.39 | 64.92±0.43 | 94.57±0.23 | 94.94±0.21 | 93.91±0.42 | 93.83±0.45 | OOM | OOM |
| SeHGNN | 66.63±0.34 | 68.21±0.32 | 50.71±0.44 | 63.41±0.47 | 94.86±0.14 | 95.24±0.13 | 93.95±0.48 | 93.87±0.50 | 55.95±0.11 | 53.99±0.18 |
| REGCN | 65.94±0.43 | 67.89±0.43 | 40.45±1.40 | 64.16±0.65 | 94.80±0.23 | 95.18±0.21 | 93.99±0.41 | 93.90±0.42 | OOM | OOM |
| SlotGAT | 64.05±0.60 | 68.64±0.33 | 49.68±1.97 | 66.83±0.30 | 94.95±0.20 | 95.31±0.19 | 93.99±0.23 | 94.06±0.22 | OOM | OOM |
| PSHGCN | 67.10±0.60 | 69.79±0.52 | 40.01±8.26 | 62.70±0.77 | 95.27±0.13 | 95.61±0.12 | 94.35±0.23 | 94.27±0.23 | 56.16±0.21 | 54.57±0.16 |
| sRGCN$_{\text{Atomic}}$ | **68.97±0.09**$^\ddagger$ | **71.20±0.17**$^\dagger$ | **55.40±1.25**$^\dagger$ | **67.32±0.66** | **95.55±0.13**$^\ddagger$ | **95.85±0.12**$^\ddagger$ | **94.36±0.22** | **94.29±0.22** | **57.35±0.12**$^\dagger$ | **55.21±0.23**$^\dagger$ |

| HGNN | Amazon | | LastFM | | PubMed | |
|---|---|---|---|---|---|---|
| | ROC-AUC | MRR | ROC-AUC | MRR | ROC-AUC | MRR |
| RGCN | 86.34±0.24 | 93.92±0.16 | 57.21±0.09 | 77.68±0.17 | 84.62±0.33 | 94.27±0.51 |
| HGT | 88.26±2.06 | 93.87±0.65 | 54.99±0.28 | 74.96±1.46 | 85.38±1.20 | 94.98±0.69 |
| SimpleHGN | 93.40±0.62 | 96.94±0.29 | 67.59±0.23 | 90.81±0.32 | 85.48±1.08 | 93.67±1.06 |
| SeHGNN | 91.67±0.94 | 95.83±0.58 | 66.59±0.62 | 88.61±1.25 | 85.86±1.11 | 95.09±0.74 |
| SlotGAT | 95.17±0.11 | 98.00±0.09 | 70.33±0.13 | 91.72±0.50 | 88.07±0.20 | 94.71±0.33 |
| PSHGCN | 94.12±0.58 | 97.93±0.46 | 69.25±0.63 | 91.19±0.51 | 87.16±1.89 | 95.01±1.26 |
| sRGCN$_{\text{Atomic}}$ | **97.85±0.07**$^\dagger$ | **99.26±0.05**$^\dagger$ | **77.10±0.17**$^\dagger$ | **93.70±0.16**$^\dagger$ | **90.11±0.19**$^\dagger$ | **96.14±0.04**$^\dagger$ |

Motivated by this perspective, we introduce a simplified variant of RGCN, denoted *sRGCN*. It preserves the RGCN structure but replaces feature transformation matrices with relation-specific scalars,

$$\boldsymbol{W}_r^{(\ell)} = \theta_r^{(\ell)}\boldsymbol{I}.$$

This design yields a minimal yet effective baseline, naturally aligned with the requirements of *atomic HINs*. Full update rules and implementation details are provided in Appendix D.

## 5 EXPERIMENTS

In this section, we conduct experiments to address the following research questions:

- **RQ1:** Do *atomic HINs* improve performance over benchmark schemas and advanced HGNNs?
- **RQ2:** How does the *entity-attribute duality* in the atomic view benefit HGNNs?
- **RQ3:** Is schema refinement truly necessary? How does it affect node-level and link-level tasks?
- **RQ4:** How large is the performance gain across schema variants when using the same HGNN? Do schemas refined on sRGCN generalize across HGNNs, and can they be further improved through subsequent refinement?
- **RQ5:** How efficient is schema refinement when optimized through a genetic algorithm?

### 5.1 EXPERIMENT SETUP

**Datasets.** We evaluate on eight heterogeneous benchmarks drawn from diverse domains, including bibliometrics, e-commerce, knowledge graphs, social networks, and biomedicine. Dataset statistics and corresponding vanilla schemas are listed in Table 4 and Table 2, respectively. Full dataset descriptions are provided in Appendix A.1. For all datasets, we perform attribute atomization on every available attribute. When datasets include initial embeddings (e.g., pretrained language model (PLM) embeddings), we treat them as numerical attributes and likewise atomize them into feature nodes. For OGBN-MAG, following Yang et al. (2023), we initialize large-scale node types with 256-dimensional random embeddings to approximate learnable ID embeddings.

**Evaluation Setting.** We follow the experimental protocols specified for each dataset by its benchmark or those commonly adopted in the literature. For schema refinement, we employ a GA with 1024 candidates and subsequently fine-tune HGNN hyperparameters on the derived optimal schema using 256 trials. Full details of experimental setup, baseline implementations, and hyperparameter space are provided in Appendix A.

Table 2: Schema refinement results on sRGCN. Edge types with underline are weighted, and with double underline are weighted and dense. Node type aliases are listed in Appendix A.1.1.

**IMDb**

| Variant | N | R | D | W | C | L | A | O | K | M |
|---|---|---|---|---|---|---|---|---|---|---|
| Vanilla | ✓ | ✓ | – | ✓ | ✓ | ✓ | – | ✓ | ✓ | – |
| Refined | ✓ | ✓ | ✓ | ✓ | ✓ | ✓ | – | – | – | – |

| Variant | M-D | M-K | M-W | M-A | M-R | M-N | M-L | M-O | M-C |
|---|---|---|---|---|---|---|---|---|---|
| Vanilla | ✓ | ✓ | – | ✓ | – | – | – | – | – |
| Refined | ✓ | ✓ | ✓ | ✓ | ✓ | – | – | – | – |

**Freebase**

| Variant | Bk | Or | Bs | Mu | Sp | Lo | Fi | Pe |
|---|---|---|---|---|---|---|---|---|
| Vanilla | ✓ | ✓ | ✓ | ✓ | ✓ | ✓ | ✓ | ✓ |
| Refined | ✓ | ✓ | ✓ | ✓ | ✓ | – | – | – |

| Variant | Bk-Bk | Bk-Pe | Bk-Or | Bk-Bs | … | Pe-Sp | Bk-Fi | Mu-Bs | Pe-Or |
|---|---|---|---|---|---|---|---|---|---|
| Vanilla | ✓ | ✓ | ✓ | ✓ | … | ✓ | ✓ | ✓ | ✓ |
| Refined | ✓ | ✓ | ✓ | ✓ | … | – | – | – | – |

**DBLP**

| Variant | V | $P_f$ | $A_f$ | P | T | $T_f$ | A |
|---|---|---|---|---|---|---|---|
| Vanilla | ✓ | ✓ | ✓ | – | – | ✓ | ✓ |
| Refined | ✓ | ✓ | – | – | – | – | – |

| Variant | P-A | P-$P_f$ | P-T | P-V | A-$A_f$ | T-$T_f$ |
|---|---|---|---|---|---|---|
| Vanilla | ✓ | – | ✓ | ✓ | ✓ | ✓ |
| Refined | ✓ | ✓ | ✓ | ✓ | – | |

**ACM**

| Variant | T | A | S | P |
|---|---|---|---|---|
| Vanilla | ✓ | – | – | – |
| Refined | ✓ | – | – | – |

| Variant | P-P | P-A | P-S | P-T |
|---|---|---|---|---|
| Vanilla | ✓ | ✓ | ✓ | ✓ |
| Refined | ✓ | ✓ | ✓ | |

**OGBN-MAG**

| Variant | F | E | Y | P | A | I |
|---|---|---|---|---|---|---|
| Vanilla | ✓ | ✓ | – | – | ✓ | ✓ |
| Refined | ✓ | ✓ | – | – | – | – |

| Variant | P-P | P-A | P-F | A-I | P-E | P-Y |
|---|---|---|---|---|---|---|
| Vanilla | ✓ | ✓ | ✓ | ✓ | – | – |
| Refined | ✓ | ✓ | ✓ | ✓ | ✓ | – |

**Amazon**

| Variant | I | P | B | R | C |
|---|---|---|---|---|---|
| Vanilla | – | ✓ | ✓ | ✓ | ✓ |
| Refined | ✓ | – | – | – | – |

| Variant | I-$I_v$ | I-$I_p$ | I-R | I-P | I-C | I-B |
|---|---|---|---|---|---|---|
| Vanilla | ✓ | ✓ | ✓ | – | – | – |
| Refined | ✓ | ✓ | ✓ | ✓ | – | |

**LastFM**

| Variant | T | A | U |
|---|---|---|---|
| Vanilla | ✓ | ✓ | ✓ |
| Refined | ✓ | – | – |

| Variant | A-T | U-U | U-A |
|---|---|---|---|
| Vanilla | ✓ | ✓ | ✓ |
| Refined | ✓ | – | – |

**PubMed**

| Variant | C | G | $D_f$ | D | S | $S_f$ | $G_f$ | $C_f$ |
|---|---|---|---|---|---|---|---|---|
| Vanilla | ✓ | ✓ | – | ✓ | ✓ | – | – | – |
| Refined | ✓ | ✓ | ✓ | ✓ | ✓ | – | – | – |

| Variant | C-S | C-C | G-S | G-G | … | C-D | D-$D_f$ | C-G | S-$S_f$ |
|---|---|---|---|---|---|---|---|---|---|
| Vanilla | ✓ | ✓ | ✓ | ✓ | … | ✓ | – | ✓ | – |
| Refined | – | – | – | – | … | – | – | – | – |

## 5.2 PERFORMANCE ON ATOMIC HINS (RQ1)

We first evaluate the effectiveness of *atomic HINs* using sRGCN across eight benchmarks for node classification and link prediction. We compare against recent state-of-the-art HGNNs under their vanilla schemas. As shown in Table 1, sRGCN on refined atomic schemas consistently outperforms advanced HGNNs. Gains are more significant on datasets with rich attributes or complex schemas (e.g., IMDb, Amazon, Freebase), where attribute atomization increases expressiveness while refinement tailors complexity. ACM is already in atomic form with a simple schema, leaving little room for improvement. Yet, LastFM benefits substantially from refinement despite also being in atomic form with an even simpler schema. On the large-scale OGBN-MAG, improvements are mainly driven by relations induced from PLM embeddings—atomized as relation nodes—together with schema refinement, showing that even numerical attributes can yield useful relational patterns.

Overall, Macro-F1 improves by up to 6.2% for node classification and ROC-AUC by an average of 4.9% for link prediction over the strongest baselines.

## 5.3 HOW THE ATOMIC VIEW BENEFITS HGNNS? (RQ2)

Table 2 summarizes the refined schemas from *atomic HINs* using sRGCN across datasets derived from the proposed GA algorithm, comparing with the *vanilla* schemas. Details on how each vanilla schema corresponds to specific selections are provided in Appendix A.1. We make the following observations:

- **Obs 1: Attribute atomization introduces meaningful relations.** Refined schemas frequently preserve relations created when attributes are converted into entities—relations absent in vanilla schemas but induced through atomization. This pattern consistently appears in IMDb, DBLP, OGBN-MAG, and Amazon, where refined schemas select edges from atomized attributes, indicating that such induced relations provide valuable semantic signals for HGNNs.

- **Obs 2: Entities can act as strong attributes.** In Amazon, for example, *item* is originally described by attributes such as *price* or *sales-rank*. Refined schemas, however, benefit from directly learning ID embeddings for *item* nodes, while still retaining attributes like *price* and *sales-rank*. Similar effects are also observed in IMDb and PubMed.

- **Obs 3: Relations from numerical attributes can be surprisingly useful.** We atomize all attributes, including numerical ones, which induce relations corresponding to dense adjacencies. Although such dense edges may appear unintuitive, many are consistently selected in refined schemas (highlighted with double underlines in Table 2). A plausible explanation is that these edges encode similarity relations through metapaths (Sun et al., 2011). For example, the paper–author–paper metapath in the citation network captures co-authorship, while the paper–

Table 3: Performance of HGNNs under different schema variants. **Vanilla**: the original benchmark schema. **Refined(HGNN)**: schema refined using the corresponding HGNN. The best results within each HGNN are in **bold**, the second-best are underlined, and global best across all HGNNs are highlighted with **blue bold**.

| HGNN | Schema | IMDb | | Freebase | | Amazon | | OGBN-MAG | |
|---|---|---|---|---|---|---|---|---|---|
| | | Macro-F1 | Micro-F1 | Macro-F1 | Micro-F1 | ROC-AUC | MRR | Acc. (Val) | Acc. (Test) |
| sRGCN | Vanilla | 67.64±0.41 | 70.05±0.50 | 52.13±1.78 | 67.09±0.43 | 95.94±0.28 | 98.43±0.18 | 56.73±0.21 | 54.63±0.23 |
| | Refined (sRGCN) | **68.97**±**0.09** | **71.20**±**0.17** | **55.40**±**1.25** | 67.32±0.66 | **97.85**±**0.07** | **99.26**±**0.05** | **57.35**±**0.12** | **55.21**±**0.23** |
| SimpleHGN | Vanilla | 63.53±1.36 | 67.36±0.57 | 47.72±1.48 | 66.29±0.45 | 93.40±0.62 | 96.94±0.29 | OOM | OOM |
| | Refined (sRGCN) | 65.89±0.67 | 68.60±1.13 | 53.45±1.88 | 67.83±0.33 | 96.50±0.87 | 98.61±0.49 | OOM | OOM |
| | Refined (SimpleHGN) | **67.38**±**0.80** | **70.02**±**0.62** | **53.51**±**1.39** | **67.94**±**0.80** | **97.40**±**1.11** | **99.05**±**0.08** | OOM | OOM |
| PSHGCN | Vanilla | 67.10±0.60 | 69.79±0.52 | 40.01±8.26 | 62.70±0.77 | 94.12±0.58 | 97.93±0.46 | 56.61±0.11 | 54.53±0.20 |
| | Refined (sRGCN) | **67.89**±**0.56** | **69.87**±**1.04** | 45.53±3.02 | 65.66±0.25 | 96.73±0.53 | 98.79±0.23 | 57.65±0.18 | 55.28±0.19 |
| | Refined (PSHGCN) | **67.89**±**0.56** | **69.87**±**1.04** | **48.19**±**1.45** | **66.36**±**0.75** | **97.13**±**0.11** | **98.91**±**0.05** | **57.65**±**0.09** | **55.34**±**0.21** |

embedding–paper metapath in OGBN-MAG approximates paper similarity, akin to dot-product signals between embeddings.

## 5.4 IS SCHEMA REFINEMENT NECESSARY? (RQ3)

Having examined the role of *atomic HINs* and attribute atomization, we now turn to the necessity of *schema refinement*. Using the refined schemas reported in Table 2, we analyze how selectively retaining or discarding node and edge types affects performance.

- **Obs 4: Schema refinement remains important even for fully atomic schemas.** Some datasets, such as Freebase and LastFM, are already in atomic form and therefore cannot benefit from further atomization. Nevertheless, simple refinement strategies—selectively dropping node or edge types—still yield significant improvements, highlighting the value of schema selection beyond atomization alone.

- **Obs 5: Schema refinement for link prediction favors pruning relations, even target ones.** For link prediction tasks, refinement often requires more aggressive relation removal. In LastFM, dropping the user–artist relation (the prediction target) improves results, while in PubMed, discarding all edges yields the best performance. This behavior aligns with the over-smoothing effect observed by Butler et al. (2023): link prediction is more sensitive to excessive connectivity, whereas node classification typically benefits from homophily. Pruning relations that reduce connectivity can therefore lead directly to performance gains.

## 5.5 GENERALIZATION OF REFINED SCHEMAS (RQ4)

We now examine the quantitative benefits of schema refinement and its transferability across different HGNNs (Table 3). Beyond sRGCN, we evaluate two additional models: SimpleHGN, a representative attention-based HGNN, and PSHGCN, a recent precomputation-based HGNN that models SHGC without first-order approximation. Further discussion of these models and their connections to SHGC is provided in Appendix C.

- **Obs 6: Schema variants yield substantial performance differences.** Across all three HGNNs, refined schemas consistently outperform vanilla ones (first row of each HGNN), showing that schema choice can be as influential as model architecture.

- **Obs 7: Refined schemas transfer effectively across HGNNs.** Schemas discovered with sRGCN generalize well to other HGNNs (second row of each HGNN), often delivering strong results without re-optimization. A notable case is PSHGCN on OGBN-MAG, where the transferred schema achieves even better performance.

- **Obs 8: Subsequent refinement provides small but consistent gains.** Re-optimizing schemas for each HGNN (third row of each HGNN) yields additional improvements, though the margin is modest compared to the leap from vanilla to refined (sRGCN). This suggests that schemas found with sRGCN are already near-optimal for other HGNNs.

## 5.6 HOW EFFICIENT IS SCHEMA REFINEMENT AS A SEARCH PROBLEM? (RQ5)

We assess the efficiency of schema refinement by tracking performance over search trials (Figure 3). Starting from the vanilla schema, refinement converges rapidly even on relation-rich datasets. For

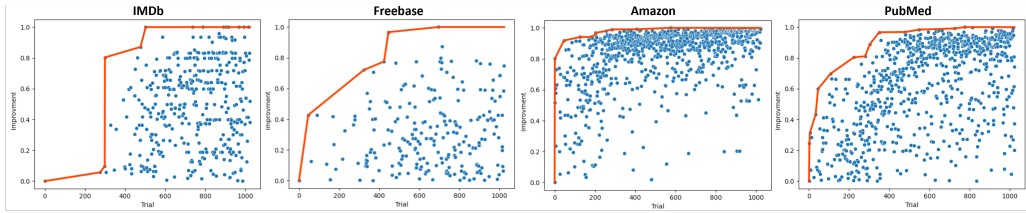

Figure 3: Performance improvement across schema search trials on four datasets.

instance, IMDb and PubMed achieve near-optimal performance within 512 trials, despite exhaustive search spaces containing up to $2^{19}$ and $2^{22}$ candidates, respectively.

## 5.7 COMPLEXITY ANALYSIS

We analyze the computational cost of the framework—including *attribute atomization*, *pre-propagation*, *type selection*, and the *search procedure*—as well as the parameter complexity induced on downstream HGNNs after schema refinement.

**Attribute atomization.** The atomization step converts node features into edges in the atomic HIN. This operation simply restructures how the data is stored, thereby incurring minimal cost. It is executed once per dataset as an offline preprocessing step.

**Pre-propagation.** Pre-propagation, used to obtain $\boldsymbol{X}(1, \ldots, 1)$ in Equation 4, can likewise be computed offline. As detailed in Appendix B.4, its cost can be upper-bounded by iteratively applying message passing on the atomic HIN at most $N$ times, where $N$ is the maximum number of relations connecting any pair of node types. In practice, $N$ is a small constant (the largest among our datasets is 4 for atomic DBLP) and is further bounded by the number of node types. As a result, the complexity of pre-propagation is $O(|\mathcal{T}||\mathcal{E}|)$, incurred once per dataset.

**Type selection.** Type selection removes all unselected relations and retains only the feature columns of the precomputed pre-propagated matrix corresponding to the selected node types. This step involves only indexing and filtering operations, and thus adds negligible computational cost.

**Effect on HGNN parameter complexity.** Schema refinement directly influences the number of relations and the feature dimensionality exposed to the HGNN. On a refined HIN, the number of selected relations is $M = \sum_{r=1}^{|\mathcal{R}|} \alpha_r$, and the input feature dimension is $D = \sum_{\tau=1}^{|\mathcal{T}|} \beta_\tau |\mathcal{V}_\tau|$, where $\mathcal{V}_\tau = \{v \in \mathcal{V} \mid \phi(v) = \tau\}$.

For HGNNs that instantiate SHGC or its first-order approximations, the parameter complexity becomes $O(M^L D)$ or $O(MLD)$, where $L$ denotes the SHGC order (or equivalently, the number of layers in first-order formulations).

**Search procedure.** Finally, the cost of schema refinement is dominated by the HGNN's training time under each candidate schema. With a search budget of $B$ trials, the overall time complexity of the search process scales linearly as $O(B)$.

## 6 CONCLUSION

We introduced the *atomic HIN*, grounded in the principle of *entity-attribute duality*, in which attributes can be atomized into entities, and entities can, in turn, serve as attributes of others. This duality maximizes expressiveness by making schema choices explicit. On this foundation, we proposed schema refinement, a systematic procedure for selecting or discarding node and edge types to yield task-specific schemas. Across eight datasets, even a simplified RGCN (sRGCN) trained on refined *atomic HINs* achieves state-of-the-art performance, with further gains from advanced HGNNs. These results establish *atomic HINs* as a powerful representation for heterogeneous graph learning and highlight schema as a central dimension of HGNN design and evaluation. Looking ahead, we hope this work provides a foundation for principled schema-aware learning, automated schema discovery, and next-generation HGNN architectures.

REPRODUCIBILITY

To ensure reproducibility and facilitate future research, we provide detailed descriptions of the experimental setup in Appendix A.3 and the hyperparameter search space in Appendix A.4. The complete source code, together with processed datasets and search configurations, is available at `https://github.com/ntuidssplab/AtomHIN`.

ACKNOWLEDGMENTS

This work was supported in part by the National Science and Technology Council and AviviD.ai under Grant 114-2622-E-002-018, and National Taiwan University and Academia Sinica under Grant NTU-AS-114L104302. Chih-Yu Wang was supported by the National Science and Technology Council under Grant 111-2628-E-001-002-MY3, 114-2221-E-001-017-MY2, and the Academia Sinica under Research Grant AS-KPQ-112-NETZ-10-A.

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

## A    EXPERIMENTAL SETUP

### A.1    DATASETS

We provide detailed descriptions of the eight heterogeneous datasets used in our experiments. For each dataset, we highlight the vanilla schema, the preprocessing conventions used in prior benchmarks, and how our atomization modifies the schema.

- **IMDb** (Lv et al., 2021): The vanilla IMDb contains four entities: *movie*, *actor*, *director*, and *keyword*. Each *movie* node has five attributes—*word*, *color*, *country*, *content rating*, and *language*. *Actor* and *director* nodes use features propagated from *word* attributes of connected *movie* nodes, while *keyword* nodes are represented by a one-hot identity matrix. Relations include *movie–actor*, *movie–director*, and *movie–keyword*. In the atomic IMDb, the attributes *word*, *color*, *country*, *content rating*, and *language* are atomized into distinct entities connected to *movie*. Under the **schema refinement framework**, the vanilla configuration selects the entities *keyword*, *word*, *color*, *country*, *content rating*, and *language*, while *movie*, *actor*, and *director* are unselected. The selected relations are *movie–actor*, *movie–director*, and *movie–keyword*. Note that *keyword* and *word* represent distinct textual levels: a *keyword* such as "coming of age" corresponds to multiple *word* tokens ("coming," "of," "age") with frequency statistics.

- **Freebase** (Lv et al., 2021): A knowledge graph comprising eight node types—*book*, *organization*, *business*, *music*, *sports*, *location*, *film*, and *people*—and thirty-six edge types connecting all possible pairs of node types. The dataset contains no attributes; all information is represented by entity identities and relations. As a result, the **atomic** and **vanilla** schemas coincide: all eight entity types are retained as nodes with unique identity embeddings, and all thirty-six relation types are preserved. Within the **schema refinement framework**, this corresponds to the configuration where all entities and all relations are selected.

- **DBLP** (Lv et al., 2021): A bibliographic network containing four entities: *author*, *paper*, *venue*, and *term*. *Term* nodes use 50-dimensional PLM features, *paper* and *author* nodes use sparse multi-hot features, and *venue* nodes are represented by a one-hot identity matrix. Relations include *author–paper*, *paper–term*, and *paper–venue*. In the atomic DBLP, the features of *paper* and *author* are atomized into entities *paper-feat* and *author-feat*, connected via *paper–paper-feat* and *author–author-feat*. The PLM feature of *term* is atomized into *term-feat* and linked through *term–term-feat*. Under the **schema refinement framework**, the vanilla configuration selects the entities *author-feat*, *venue*, *term-feat*, and *paper-feat*, and retains the relations *author–paper*, *paper–term*, and *paper–venue*.

- **ACM** (Lv et al., 2021): A citation network containing four node types: *paper*, *subject*, *author*, and *term*. *Term* nodes are represented by a one-hot identity matrix, and *paper*, *subject*, and *author* nodes are represented by propagated *term* features through *term–paper*, *term–paper–subject*, and *term–paper–author*. No explicit attributes are available for atomization. Under the **schema refinement framework**, the vanilla configuration selects only the entity *term* and all relations.

- **OGBN-MAG** (Hu et al., 2020a): A large-scale academic graph for venue classification. In its vanilla form, it contains four node types: *paper*, *author*, *institution*, and *field-of-study*. *Paper* nodes are represented by 128-dimensional PLM embeddings (mean of connected term embeddings) and include a *year* attribute used only for data splitting: pre-2018 (train), 2018 (validation), and post-2018 (test). Other node types have no attributes and are initialized with 256-dimensional random vectors as in (Yang et al., 2023). In the atomic OGBN-MAG, the PLM embeddings are atomized into *embedding* nodes linked via *paper–embedding*, and the *year* attribute is atomized into nodes connected by *paper–year*, constructed only for training papers to avoid data leakage. Under the **schema refinement framework**, the vanilla configuration selects the node types *embedding*, *author*, *institution*, and *field-of-study*, while *paper* and *year* are unselected. The selected

Table 4: Dataset statistics for benchmark and atomic HIN.

| Dataset | Target (type / etype) | #Classes | Benchmark (vanilla) | | | | Atomic HIN | | | | |
|---|---|---|---|---|---|---|---|---|---|---|---|
| | | | #Nodes | #Node-Types | #Edges | #Edge-Types | #Nodes | #Node-Types | #Edges | #Edge-Types | Dense Adj. |
| *Node Classification (NC)* | | | | | | | | | | | |
| IMDb | Movie | 5 | 21,420 | 4 | 43,321 | 6 | 24,909 | 10 | 94,384 | 9 | 4,932×16 |
| Freebase | Book | 7 | 180,098 | 8 | 1,057,688 | 36 | 180,098 | 8 | 1,057,688 | 36 | — |
| DBLP | Author | 4 | 26,128 | 4 | 119,783 | 3 | 30,743 | 7 | 263,623 | 6 | 7,723×50 |
| ACM | Paper | 3 | 10,942 | 4 | 279,221 | 4 | 10,942 | 4 | 279,221 | 4 | — |
| OGBN-MAG | Paper | 349 | 1,939,743 | 4 | 21,111,007 | 4 | 1,939,879 | 6 | 21,740,578 | 6 | 736,389×128 |
| *Link Prediction (LP)* | | | | | | | | | | | |
| Amazon | item-item | — | 10,099 | 1 | 121,470 | 2 | 11,256 | 5 | 151,729 | 6 | 10,099×2 |
| LastFM | artist-user | — | 20,612 | 3 | 111,796 | 3 | 20,612 | 3 | 111,796 | 3 | — |
| PubMed | disease-disease | — | 63,109 | 4 | 233,047 | 10 | 63,909 | 8 | 233,047 | 14 | 200×13,561 |

relations include *paper–author*, *author–institution*, *paper–field-of-study*, and *paper–paper*, with *paper–embedding* and *paper–year* unselected. Note that for large-scale node types (*author*, *institution*, *field-of-study*, *paper*), random embeddings are applied following prior practice, while smaller node types (*embedding*, *year*) use full identity matrices.

- **Amazon** (Cen et al., 2019): An e-commerce graph whose *vanilla form* contains a single node type, *item*, and two relation types: *co-view* and *co-purchase*. In the atomic Amazon, new node types are atomized from item attributes, including *price*, *sales-rank*, *category*, and *brand*, along with their connecting relations. Under the **schema refinement framework**, the vanilla configuration selects the node types *price*, *sales-rank*, *category*, and *brand*, with *item* unselected. All relations introduced by atomization are unselected. Note that the *price* attribute is one-dimensional; direct normalization collapses values to $\pm 1$. A dummy column equal to the mean absolute price is appended before normalization to ensure non-trivial edge weights for *item–price*.

- **LastFM** (Cantador et al., 2011): A music network containing the node types *user*, *artist*, and *tag*, targeting *user–artist* link prediction. All nodes lack attributes and are represented by identity matrices. Since no attributes are available in the benchmark, the graph is already atomic. Under the **schema refinement framework**, the vanilla configuration selects all node types and relations.

- **PubMed** (Yang et al., 2020): A biomedical knowledge graph whose vanilla form contains four node types: *disease*, *gene*, *chemical*, and *species*, connected by ten relations. All nodes have 256-dimensional PLM embeddings, which are often replaced by learnable identity embeddings in practice. In the atomic PubMed, these embeddings are atomized into feature nodes *disease-feat*, *gene-feat*, *chemical-feat*, and *species-feat*, with corresponding relations to their base nodes. Under the **schema refinement framework**, the vanilla configuration selects the node types *disease*, *gene*, *chemical*, and *species*, as baselines typically use identity embeddings. All original relations not induced through atomization are selected.

### A.1.1 ALIASES FOR NODE TYPES

Aliases for node types used in Table 2:

- **IMDb:** N = numerical, R = content rating, D = director, W = word, C = country, L = language, A = actor, O = color, K = keyword.

- **Freebase:** Bk = book, Or = organization, Bs = business, Mu = music, Sp = sports, Lo = location, Fi = film, Pe = people.

- **DBLP:** A = author, P = paper, V = venue, T = term, $A_f$ = author-feature, $P_f$ = paper-feature, $T_f$ = term-feature.

- **ACM:** P = paper, A = author, S = subject, T = term.

- **Amazon:** I = item, P = price, B = brand, R = sales-rank, C = category.

- **LastFM:** U = user, A = artist, T = tag.

- **PubMed:** D = disease, G = gene, C = chemical, S = species, $D_f$ = disease-feature, $G_f$ = gene-feature, $C_f$ = chemical-feature, $S_f$ = species-feature.

- **OGBN-MAG:** P = paper, A = author, I = institution, F = field-of-study, E = term-embedding, Y = year.

## A.2 Baselines HGNNs

We briefly describe the HGNN baselines used in our experiments:

- **XGBoost** (Chen & Guestrin, 2016): A strong non-neural baseline for tabular learning. We apply it to node classification datasets with target-node features, providing a feature-based comparison independent of graph structure.

- **GTN** (Yun et al., 2019): Learns soft selections over edge types to construct composite relations, generating metapath-based adjacency structures on which a GCN is applied. This enables dynamic metapath learning rather than relying on manually defined schemas.

- **RGCN** (Schlichtkrull et al., 2018): One of the earliest HGNNs, extending GCNs to relational graphs by introducing relation-specific feature transformations. Each edge type is assigned a distinct projection matrix, enabling type-aware message passing.

- **HGT** (Hu et al., 2020b): An early attention-based HGNN. It models heterogeneity through relation-specific attention with node-type-aware key/query projections, combined with learnable relation-specific priors to capture the varying importance of different relations.

- **SimpleHGN** (Lv et al., 2021): A simple extension of GAT without heavily modeling heterogeneity. It incorporates relation-aware information through relation embeddings, achieving competitive performance.

- **HINormer** (Mao et al., 2023): Adapts Graph Transformers to HINs by combining them with GCN-style propagation. It first encodes local structure through neighborhood aggregation, then applies a heterogeneous relation encoder to model relation-specific information.

- **SeHGNN** (Yang et al., 2023): Extends SGC-style precomputation to HINs, enabling efficient precomputation-based training. This design allows mini-batch learning without information loss from subgraph sampling or GPU memory overhead, making it well-suited for large-scale settings.

- **RE-GCN** (Wang et al., 2023): Learns relation embeddings for heterogeneous graphs with gradient scaling, enabling GCN-style aggregation over soft composite relation subgraphs and improving its ability to handle heterogeneity.

- **SlotGAT** (Zhou et al., 2023): A GAT-based model for heterogeneous graphs. It conducts message passing separately across node-type-specific "slots," preserving distinct semantics in different feature spaces.

- **PSHGCN** (He et al., 2024): A spectral HGNN that models heterogeneous graphs under positive semi-definite constraints on spectral filters. Its framework also extends to precomputation-based settings, enabling scalable and efficient training on large-scale HINs.

## A.3 Evaluation Setting

For node classification datasets (IMDb, Freebase, DBLP, ACM, OGBN-MAG), we follow the official splits and evaluation metrics provided by each benchmark. On OGBN-MAG, we report classification accuracy on both validation and test sets, following the official protocol, and present the mean and standard deviation over 10 runs. On the other datasets, we report both Macro-F1 and Micro-F1, averaged over 5 runs.

For link prediction datasets (Amazon, LastFM, PubMed), the task is cast as binary classification on node pairs. We evaluate with ROC-AUC and MRR, using negative samples drawn from 2-hop neighbors of each positive pair, following the protocol of Lv et al. (2021). All datasets use official splits except PubMed, which was originally introduced by Yang et al. (2020). We found that its official split suffers from a severe distribution shift, so we re-split the dataset (with the same ratio). All experiments are conducted on a single RTX A6000 GPU with 48GB memory or smaller GPUs.

## A.4 Hyperparameter Setting

As discussed in Section 4.2, we treat schema parameters and model depth as hyperparameters. Hyperparameter optimization is carried out using `optuna` (Akiba et al., 2019). We adopt a two-stage tuning process: (1) schema tuning with a budget of 1024 trials, followed by (2) HGNN hyperparameter tuning with 256 trials.

**Schema tuning.** We use a GA (Deb et al., 2002) with default parameters, except that mutation probability is set to $1/8$ for datasets with more than eight schema parameters. The search is initialized from the vanilla schema configuration and explores 1024 candidate points.

**HGNN hyperparameter tuning.** We apply the Tree-structured Parzen Estimator (TPE) (Ozaki et al., 2020) implemented in `optuna`, using default parameters. For each model, we define the following search spaces:

- **sRGCN:** number of layers, hidden dimension, number of heads, dropout rate, edge dropout (or input dropout for precomputation-based setting on OGBN-MAG), learning rate, weight decay, decoder MLP layers.

- **SimpleHGN (Lv et al., 2021):** number of layers, hidden dimension, number of heads, dropout rate, edge dropout, learning rate, weight decay.

- **PSHGCN (He et al., 2024):** number of layers/hops, hidden dimension, embedding dimension, dropout rate, input dropout, learning rate, weight decay.

## B    DETAILS OF PRE-PROPAGATION FEATURE INITIALIZATION

This section provides further motivation and proofs for the lemmas stated in Section 4.3, along with additional technical details.

### B.1    PROOF OF LEMMA 4.1

**Lemma** (Independence of Selections). *With pre-propagation, for any node type $\tau_i$ and any selected type $\tau_j$ with $\beta_{\tau_j} = 1$, $\hat{\boldsymbol{I}}_{\tau_i} \bar{\boldsymbol{Z}} \hat{\boldsymbol{I}}_{\tau_j} \neq \boldsymbol{0}$. Hence, node-type selection is independent of edge-type selection: even if all edges incident to $\tau_j$ are removed, no dependency arises.*

*Proof.* Expanding the SHGC polynomial yields

$$\bar{\boldsymbol{Z}} = \boldsymbol{H}(\alpha_1 \boldsymbol{S}_1, \ldots, \alpha_{|\mathcal{R}|} \boldsymbol{S}_{|\mathcal{R}|}) \boldsymbol{X}(\beta_1, \ldots, \beta_{|\mathcal{T}|})$$

$$= \theta_0 \boldsymbol{X}(\beta_1, \ldots, \beta_{|\mathcal{T}|}) + \sum_{\ell=1}^{L} \sum_{r_1, \ldots, r_\ell} \theta_{r_1, \ldots, r_\ell} (\boldsymbol{S}_{r_1} \cdots \boldsymbol{S}_{r_\ell}) \boldsymbol{X}(\beta_1, \ldots, \beta_{|\mathcal{T}|}). \tag{5}$$

The first term guarantees that $\hat{\boldsymbol{I}}_{\tau_i} \boldsymbol{X}(\beta_1, \ldots, \beta_{|\mathcal{T}|}) \hat{\boldsymbol{I}}_{\tau_j} \neq \boldsymbol{0}$ for all node types $\tau_i$ whenever $\beta_{\tau_j} = 1$, independent of the edge-type selectors $\{\alpha_r\}_r$. Hence, even if all incident relations of $\tau_j$ are pruned, its pre-propagated identity remains accessible to all other node types, confirming independence.    □

### B.2    PROOF OF LEMMA 4.2

**Lemma** (Neutrality of Pre-propagation). *Consider an SHGC with row-normalized adjacencies $\boldsymbol{S}_r = \tilde{\boldsymbol{A}}_r$ as shift operators and convolution order $L$ sufficiently large. If raw identity features $\boldsymbol{X}_0(1, \ldots, 1) = \boldsymbol{I}$ are replaced with pre-propagated features $\boldsymbol{X}(1, \ldots, 1)$, the effect is merely a reparameterization of the filter coefficients $\{\theta_{r_1, \ldots, r_\ell}\}_{r_1, \ldots, r_\ell}$. Hence, pre-propagation does not alter the expressive power of the model.*

*Proof.* The SHGC representation with identity features $\boldsymbol{I}$ can be written as

$$\tilde{\boldsymbol{Z}} = \theta_0 \boldsymbol{I} + \sum_{\ell=1}^{L} \sum_{r_1, \ldots, r_\ell} \theta_{r_1, \ldots, r_\ell} (\tilde{\boldsymbol{A}}_{r_1} \cdots \tilde{\boldsymbol{A}}_{r_\ell}) \boldsymbol{I}.$$

After applying pre-propagation, the identity features are replaced by

$$\boldsymbol{X}(1, \ldots, 1) = (\boldsymbol{I} + \sum_{\tau_i \neq \tau_j} \tilde{\boldsymbol{A}}_{\langle \tau_i, *, \tau_j \rangle}) \boldsymbol{I},$$

yielding

$$\tilde{\boldsymbol{Z}}' = \left(\theta_0 \boldsymbol{I} + \sum_{\ell=1}^{L} \sum_{r_1,\ldots,r_\ell} \theta_{r_1,\ldots,r_\ell} \left(\tilde{\boldsymbol{A}}_{r_1} \cdots \tilde{\boldsymbol{A}}_{r_\ell}\right)\right) \boldsymbol{X}(1,\ldots,1)$$

$$\approx \left(\theta_0 \boldsymbol{I} + \sum_{\ell=1}^{L} \sum_{r_1,\ldots,r_\ell} \theta'_{r_1,\ldots,r_\ell} \left(\tilde{\boldsymbol{A}}_{r_1} \cdots \tilde{\boldsymbol{A}}_{r_\ell}\right)\right) \boldsymbol{I}, \tag{6}$$

where the updated coefficients $\{\theta'_{r_1,\ldots,r_\ell}\}$ result from expanding the product with $\boldsymbol{X}(1,\ldots,1)$. Since the operator span $\{\tilde{\boldsymbol{A}}_{r_1} \cdots \tilde{\boldsymbol{A}}_{r_\ell}\}_{r_1,\ldots,r_\ell}$ remains unchanged given sufficiently large $L$, pre-propagation only reparameterizes the convolution coefficients without altering the expressive power of the model. □

## B.3 Generalized union aggregation.

We first demonstrate pre-propagation using row-normalized adjacencies $\tilde{\boldsymbol{A}}_r$, which corresponds to mean or weighted-sum pooling:

$$\bar{\boldsymbol{x}}_i = \sum_{j \in \mathcal{N}_i^{(r)}} \tilde{A}_r[i,j]\, \boldsymbol{x}_j, \tag{7}$$

where $\mathcal{N}_i^{(r)}$ denotes the neighbors of node $i$ under relation $r$.

In practice, however, union-based aggregation is often used to initialize features, as in Lv et al. (2021). In the unweighted case, this is equivalent to an element-wise maximum across neighbors:

$$\bar{x}_i[k] = \max_{j \in \mathcal{N}_i^{(r)}} x_j[k]. \tag{8}$$

We extend this to a generalized union operator that performs element-wise maximization while ignoring sign. This variant naturally handles weighted adjacencies and empirically yields more stable performance when used for pre-propagation:

$$\bar{x}_i[k] = A_r[i,t]\, x_t[k], \qquad t = \arg \max_{j \in \mathcal{N}_i^{(r)}} |A_r[i,j]\, x_j[k]|. \tag{9}$$

Here the chosen neighbor $t$ is the one with the maximum (magnitude) contribution on dimension $k$. Overall, we find that union-based pre-propagation tends to yield more consistent performance across datasets compared to mean or weighted-sum pooling with $\tilde{\boldsymbol{A}}_r$.

## B.4 Implementation.

Pre-propagation is performed before type selection. This procedure is independent of edge-type selection and thus needs to be precomputed only once per dataset.

The computation proceeds independently for each node type. For a given node type $\tau_j$, we initialize its feature matrix as $\boldsymbol{X}_{\tau_j} = \hat{\boldsymbol{I}}_{\tau_j}$, i.e., identity features for nodes of type $\tau_j$ and zeros elsewhere. Propagation is then applied along any relation whose source node type already has non-empty features and whose destination node type is still empty. This process is repeated iteratively until all node types have received non-empty features, yielding the pre-propagated initialization $\boldsymbol{X}(1,\ldots,1)\hat{\boldsymbol{I}}_{\tau_j}$ for node type $\tau_j$.

The full procedure is summarized in Algorithm 1.

## C SHGC Perspective on HGNNs

This appendix provides detailed case analyses of how representative HGNNs relate to the Spectral Heterogeneous Graph Convolution (SHGC) formulation introduced in Equation 1. We show that several widely used models can be interpreted as first-order approximations of SHGC under different parameter-sharing schemes.

---

**Algorithm 1** Pre-propagation of features

---

**Input:** $\{\tilde{A}_r\}_r$: row-normalized adjacency matrices of relations in the heterogeneous graph
**Input:** $\{X_\tau\}_{\tau=1}^{|\mathcal{T}|}$: raw feature matrices for each node type
**Output:** $\{X'_\tau\}_{\tau=1}^{|\mathcal{T}|}$: pre-propagated feature matrices
 1: **procedure** PREPROPAGATE($\{\tilde{A}_r\}_r$, $\{X_\tau\}_{\tau=1}^{|\mathcal{T}|}$)
 2:     Initialize $\{X'_\tau\}_{\tau=1}^{|\mathcal{T}|} \leftarrow \{\mathbf{0}\}_{\tau=1}^{|\mathcal{T}|}$
 3:     **for** $\tau_j \leftarrow 1$ **to** $|\mathcal{T}|$ **do**
 4:         Construct $\{X_{\tau_i}^{(\tau_j)}\}_{\tau_i=1}^{|\mathcal{T}|}$ where $X_{\tau_j}^{(\tau_j)} \leftarrow X_{\tau_j}$ and $X_{\tau_i}^{(\tau_j)} \leftarrow \mathbf{0}$ for $\tau_j \neq \tau_i$
 5:         $\{Z_{\tau_i}\}_{\tau_i=1}^{|\mathcal{T}|} \leftarrow$ PROPAGATEFEATURES($\{\tilde{A}_r\}_r$, $\{X_{\tau_i}^{(\tau_j)}\}_{\tau_i=1}^{|\mathcal{T}|}$)
 6:         **for** $\tau_i \leftarrow 1$ **to** $|\mathcal{T}|$ **do**
 7:             $X'_{\tau_i} \leftarrow X'_{\tau_i} + Z_{\tau_i}$
 8:         **end for**
 9:     **end for**
10:     **return** $\{X'_\tau\}_{\tau=1}^{|\mathcal{T}|}$

11:     **function** PROPAGATEFEATURES($\{\tilde{A}_r\}_r$, $\{X_\tau\}_{\tau=1}^{|\mathcal{T}|}$)
12:         **while** $\exists \tau_i$ such that $X_{\tau_i} = \mathbf{0}$ **do**
13:             **for all** $\tilde{A}_r \in \{\tilde{A}_r\}_r$ **do**
14:                 $(\tau_s, \tau_t) \leftarrow$ source and target node types of relation $r$
15:                 **if** $X_{\tau_s} \neq \mathbf{0}$ and $X_{\tau_t} = \mathbf{0}$ **then**
16:                     $X_{\tau_t} \leftarrow \tilde{A}_r X_{\tau_s}$
17:                 **end if**
18:             **end for**
19:         **end while**
20:         **return** $\{X_\tau\}_{\tau=1}^{|\mathcal{T}|}$
21:     **end function**
22: **end procedure**

---

Table 5: Representative HGNNs interpreted under SHGC. Approximation indicates whether the model corresponds to a first-order approximation of SHGC or models SHGC directly. Shift operators: $\tilde{A}_r$ denotes relation-specific row-normalization, and $\hat{A}_r$ denotes cross-type row-normalization.

| Model | Approximation | Shift Operator | Parameter Sharing |
|---|---|---|---|
| RGCN (Schlichtkrull et al., 2018) | First-order | $\tilde{A}_r$ | $W_r^{(\ell)} = \sum_b \theta_{r,b} W_b^{(\ell)}$ |
| sRGCN (ours) | First-order | $\tilde{A}_r$ | $W_r^{(\ell)} = \theta_r^{(\ell)} I$ |
| GTN (Yun et al., 2019) | First-order | $\tilde{A}_r$ | $W_r^{(\ell)} = \theta_r^{(\ell)} I$ |
| SimpleHGN (Lv et al., 2021) | First-order | $\hat{A}_r$ | $W_r^{(\ell)} = \theta_r^{(\ell)} W^{(\ell)}$ |
| SeHGNN (Yang et al., 2023) | – | $\tilde{A}_r$ | $W_{r_1,\dots,r_\ell}$ |
| PSHGCN (He et al., 2024) | – | $\tilde{A}_r$ | $W_{r_1,\dots,r_\ell} = \theta_{r_1,\dots,r_\ell} W$ |

Formally, extending SHGC (Equation 1) to multi-channel features $X \in \mathbb{R}^{|\mathcal{V}| \times d}$ yields

$$Z = IXW_0 + \sum_{\ell=1}^{L} \sum_{r_1,\dots,r_\ell} S_{r_1} S_{r_2} \cdots S_{r_\ell} X W_{r_1,\dots,r_\ell}, \tag{10}$$

where $W_{r_1,\dots,r_\ell} \in \mathbb{R}^{d \times d'}$ are generalized filter coefficients for multi-channel features and $S_r$ denotes the shift operator associated with relation $r$.

### C.1 FIRST-ORDER APPROXIMATIONS

**Relational GCN (RGCN).** RGCN (Schlichtkrull et al., 2018) extends GCN (Kipf & Welling, 2017) to heterogeneous graphs. Following GCN, which adopts a first-order approximation of spectral convolution (i.e., $L = 1$) and realizes higher-order convolution by stacking multiple layers,

RGCN can likewise be viewed as a first-order approximation of SHGC:

$$\boldsymbol{H}^{(\ell+1)} = \boldsymbol{I}\boldsymbol{H}^{(\ell)}\boldsymbol{W}_0^{(\ell)} + \sum_{r=1}^{|\mathcal{R}|} \tilde{\boldsymbol{A}}_r \boldsymbol{H}^{(\ell)}\boldsymbol{W}_r^{(\ell)},$$

where $\tilde{\boldsymbol{A}}_r$ denotes the normalized adjacency matrix for relation $r$.[2]

Even with this first-order approximation, the number of parameters $\{\boldsymbol{W}_r^{(\ell)} \in \mathbb{R}^{d \times d'}\}_{r,\ell}$ can still be prohibitive for relation-rich graphs. To address this, RGCN introduces a basis-decomposition scheme for parameter sharing:

$$\boldsymbol{W}_r^{(\ell)} = \sum_{b=1}^{B} \theta_{r,b}\,\boldsymbol{W}_b,$$

where $B \in \mathbb{N}$ is the number of basis transformations, $\boldsymbol{W}_b \in \mathbb{R}^{d \times d'}$ are shared across all relations, and $\theta_{r,b} \in \mathbb{R}$ are relation-specific coefficients.

**Graph Transformer Network (GTN).** GTN (Yun et al., 2019) learns weighted combinations of adjacency matrices to construct meta-relations, producing a meta-path adjacency $\boldsymbol{A}_P$ on which a GCN can be applied.

Formally, $\boldsymbol{A}_P$ is learned for an arbitrary meta-path of length at most $L$ as

$$\boldsymbol{A}_P = \prod_{\ell=1}^{L} \left( \theta_0^{(\ell)}\boldsymbol{I} + \sum_{r=1}^{|\mathcal{R}|} \theta_r^{(\ell)}\boldsymbol{A}_r \right),$$

where self-loops are included to allow paths shorter than $L$ hops. A GCN is then applied using the row-normalized adjacency $\tilde{\boldsymbol{A}}_P$ as the propagation operator:

$$\boldsymbol{Z} = (\boldsymbol{I} + \tilde{\boldsymbol{A}}_P)\boldsymbol{X}\boldsymbol{W}.$$

Expanding $\tilde{\boldsymbol{A}}_P$ shows that $\boldsymbol{Z}$ can be written as

$$\boldsymbol{Z} = \theta_0'\boldsymbol{I}\boldsymbol{X}\boldsymbol{W} + \sum_{\ell=1}^{L} \sum_{r_1,\ldots,r_\ell} \theta_{r_1,\ldots,r_\ell}' \tilde{\boldsymbol{A}}_{r_1} \cdots \tilde{\boldsymbol{A}}_{r_\ell}\,\boldsymbol{X}\boldsymbol{W},$$

where each coefficient $\theta_{r_1,\ldots,r_\ell}'$ is a product of up to $L$ scalars $\{\theta_r^{(\ell)}\}_{r,\ell}$. For example, $\theta_0' = 1 + \prod_{\ell=1}^{L} \theta_0^{(\ell)}$ corresponds to the identity term.

This expansion demonstrates that GTN is another example of a first-order approximation of SHGC, but with stronger parameter sharing:

$$\boldsymbol{W}_r^{(\ell)} = \theta_r^{(\ell)}\boldsymbol{I}.$$

In practice, GTN further introduces multiple sets of scalars $\theta_r^{(\ell,b)}$ for $b = 1, \ldots, B$ to capture diverse meta-relational structures. We omit this extension here for clarity.

**SimpleHGN.** Attention-based HGNNs, such as SimpleHGN (Lv et al., 2021), extend GAT (Veličković et al., 2018) by introducing relation-aware attention through a relation-specific bias in the attention mechanism.

Empirically, attention models behave similarly to spectral graph convolution with row-normalization (Chen et al., 2020). Assuming further that the relation-specific bias acts effectively as a multiplicative scalar, SimpleHGN can be interpreted as a first-order SHGC approximation with parameter-sharing as:

$$\boldsymbol{W}_r^{(\ell)} = \theta_r^{(\ell)}\boldsymbol{W}^{(\ell)}, \tag{11}$$

where $\boldsymbol{W}^{(\ell)} \in \mathbb{R}^{d \times d'}$ is a shared projection and $\theta_r^{(\ell)} \in \mathbb{R}$ is a relation-specific scalar.

---

[2]RGCN considers different normalization choices depending on the task: relation-specific normalization for node-level tasks, and cross-relation normalization for link-level tasks. Here, $\tilde{\boldsymbol{A}}_r$ corresponds to the relation-specific normalization.

Concretely, SimpleHGN incorporates edge-type information by introducing a learnable relation embedding $\boldsymbol{w}_r \in \mathbb{R}^{d_{\mathrm{re}}}$ (with $d_{\mathrm{re}}$ a hyperparameter) and a projection $\boldsymbol{W}_{\mathrm{rel}}$. For an edge $(i, j)$ of type $r = \psi(\langle i, j \rangle)$, the modified attention score is

$$\hat{\alpha}'_{i,j} = \boldsymbol{a}^\top \big[\, \boldsymbol{W}^\top \boldsymbol{h}_i \parallel \boldsymbol{W}^\top \boldsymbol{h}_j \parallel \boldsymbol{W}_{\mathrm{rel}}^\top \boldsymbol{w}_r \,\big]$$
$$= \hat{\alpha}_{i,j} + \boldsymbol{a}_2^\top \boldsymbol{W}_{\mathrm{rel}}^\top \boldsymbol{w}_r,$$

where $\psi : \mathcal{V} \times \mathcal{V} \to \mathcal{R}$ is the edge-type indicator, $\hat{\alpha}_{i,j} = \boldsymbol{a}_1^\top [\boldsymbol{W}^\top \boldsymbol{h}_i \parallel \boldsymbol{W}^\top \boldsymbol{h}_j]$ is the vanilla GAT attention score, and $\boldsymbol{a} = \boldsymbol{a}_1 \parallel \boldsymbol{a}_2$. (For readability, we omit the layer index $(\ell)$ in this part; it is reintroduced in the update equations below.)

Although this formulation is an intuitive extension of GAT, the relation-specific term $\boldsymbol{a}_2^\top \boldsymbol{W}_{\mathrm{rel}}^\top \boldsymbol{w}_r$ can be reparametrized by a single scalar $\theta_r$, yielding

$$\hat{\alpha}'_{i,j} \;=\; \hat{\alpha}_{i,j} + \theta_r.$$

Just as GTN introduces multiple scalar coefficients per relation, SimpleHGN extends to use multi-head attention, each head learning its own $\theta_{r,b}$.

Following the empirical observation that attention resembles row-normalized spectral convolution (Chen et al., 2020), we may approximate the relational bias as multiplicative scalar edge weights $\hat{\theta}_r^{(\ell)}$. This yields the SimpleHGN update rule

$$\boldsymbol{h}_i^{(\ell+1)} = \sum_{r=1}^{|\mathcal{R}|} \sum_{j \in \mathcal{N}_i^{(r)}} \hat{\theta}_r^{(\ell)} \boldsymbol{W}^{(\ell)\top} \boldsymbol{h}_j^{(\ell)} \;+\; \boldsymbol{W}_0^{(\ell)\top} \boldsymbol{h}_i^{(\ell)},$$

where $\mathcal{N}_i^{(r)}$ denote the neighbors of $i$-th node via edge type $r$. Or equivalently in matrix form,

$$\boldsymbol{H}^{(\ell+1)} = \sum_{r=1}^{|\mathcal{R}|} \hat{\theta}_r^{(\ell)} \hat{\boldsymbol{A}}_r \boldsymbol{H}^{(\ell)} \boldsymbol{W}^{(\ell)} \;+\; \boldsymbol{I} \boldsymbol{H}^{(\ell)} \boldsymbol{W}_0^{(\ell)},$$

where $\hat{\boldsymbol{A}}_r$ is the cross-type row-normalized adjacency matrix for relation $r$, approximating the softmax normalization of GAT. This shows the parameter-sharing introduced earlier in Equation 11.

## C.2 BEYOND FIRST-ORDER APPROXIMATIONS

While many HGNNs adopt first-order approximations, several recent works learn SHGC more directly without truncation. These models are often precomputation-based, computing $(\boldsymbol{S}_{r_1} \boldsymbol{S}_{r_2} \cdots \boldsymbol{S}_{r_\ell})\boldsymbol{X}$ offline. Such precomputation is advantageous when the number of relations is moderate, as it enables efficient mini-batch training: precomputed features $(\boldsymbol{S}_{r_1} \boldsymbol{S}_{r_2} \cdots \boldsymbol{S}_{r_\ell})\boldsymbol{X}$ can be sliced directly without subgraph sampling, thereby avoiding the trade-off between information loss (few hops or few sampled neighbors) and GPU memory overhead.

**SeHGNN.** SeHGNN (Yang et al., 2023) is one of the earliest and most representative precomputation-based HGNNs, extending SGC (Wu et al., 2019) to heterogeneous graphs. It directly parameterizes spectral filters of the form $\boldsymbol{S}_{r_1} \boldsymbol{S}_{r_2} \cdots \boldsymbol{S}_{r_\ell}$, with $\boldsymbol{S}_r = \tilde{\boldsymbol{A}}_r$ the row-normalized adjacency of relation $r$, and learns a projection $\boldsymbol{W}_{r_1,\dots,r_\ell} \in \mathbb{R}^{d \times d'}$ for each such sequence.

Thus, SeHGNN follows Equation 10 exactly, while further applying a Transformer-based semantic fusion followed by an MLP prediction head. The semantic fusion can equivalently be viewed as part of a general decoder $Y = \mathrm{MLP}(\boldsymbol{Z})$, since the metapath sequences (i.e., polynomial terms) form a fixed-length, orderless set.

**PSHGCN.** PSHGCN (He et al., 2024) is another HGNN that directly learning the coefficients $\theta_0, \theta_{r_1,\dots,r_\ell}$ of SHGC (as in Equation 1) without first-order approximation, while additionally constraining the polynomial coefficients to be positive semi-definite, ensuring the optimization objective is convex.

PSHGCN also adopts stronger parameter sharing:

$$\boldsymbol{W}_0 = \theta_0 \boldsymbol{W}, \qquad \boldsymbol{W}_{r_1,\dots,r_\ell} = \theta_{r_1,\dots,r_\ell} \boldsymbol{W},$$

where $\boldsymbol{W} \in \mathbb{R}^{d \times d'}$ is a node-type-specific projection. Since heterogeneous graphs often have features in different spaces and dimensions, $\boldsymbol{W}$ decomposes as $\boldsymbol{X}\boldsymbol{W} = \sum_{\tau=1}^{|\mathcal{T}|} \boldsymbol{X}_\tau \boldsymbol{W}_\tau$, where $\boldsymbol{X} = \bigoplus_{\tau=1}^{|\mathcal{T}|} \boldsymbol{X}_\tau$ is the direct sum of features across node types and $\boldsymbol{W}_\tau$ is the corresponding block of $\boldsymbol{W}$ applied to type $\tau$.

### C.3 SUMMARY

The above analyses show that representative HGNNs can all be interpreted within the SHGC formulation, differing mainly in (i) the order of approximation and (ii) the extent of parameter sharing. This perspective highlights two key insights: (1) representative models such as RGCN, GTN, and SimpleHGN correspond to first-order SHGC approximations with different parameter-sharing schemes, while SeHGNN and PSHGCN directly instantiate higher-order SHGC; (2) these results provide theoretical justification that atomic HINs are broadly compatible with diverse HGNN designs.

Table 5 provides a concise taxonomy of representative HGNNs under the SHGC lens.

## D  SRGCN: SIMPLIFIED RELATIONAL GCN

As discussed in Section 4.5, we propose a simplified variant of RGCN under the strongest parameter sharing, where relation-specific weights collapse to scalars:

$$\boldsymbol{W}_r^{(\ell)} = \theta_r^{(\ell)} \boldsymbol{I}.$$

**Propagation rule.**    The update rule of sRGCN is

$$\boldsymbol{H}^{(\ell+1)} = \tilde{\theta}_0^{(\ell)} \boldsymbol{H}^{(\ell)} + \sum_{r=1}^{|\mathcal{R}|} \tilde{\theta}_r^{(\ell)} \tilde{\boldsymbol{A}}_r \boldsymbol{H}^{(\ell)} + \boldsymbol{H}^{(\ell)}, \tag{12}$$

with initial features $\boldsymbol{H}^{(0)} = \boldsymbol{X}\boldsymbol{W}$. Here $\tilde{\theta}_0^{(\ell)}, \{\tilde{\theta}_r^{(\ell)}\}$ are relation coefficients normalized through a softmax, following Yun et al. (2019), for numerical stability.

**Multi-head extension.**    Following Yun et al. (2019); Lv et al. (2021), we employ a multi-head scheme, where multiple sets of coefficients $\{\theta_r^{(\ell,b)}\}_{r,\ell,b}$ are learned in parallel, thereby increasing model capacity.

**Training enhancements.**    We incorporate model-agnostic techniques commonly adopted in advanced HGNNs, including: (i) L2-normalization of output logits (Lv et al., 2021), (ii) edge dropout (or input dropout in the precomputation setting), and (iii) residual connections across layers.

**Decoders.**    Following existing practices, we apply an MLP decoder to the learned embeddings for node classification tasks. For link prediction tasks, we use either a dot-product decoder or Dist-Mult (Yang et al., 2014).

**Precomputation setting.**    Following the idea of precomputation-based HGNNs (Yang et al., 2023; He et al., 2024), sRGCN can be further simplified by removing intermediate non-linear activations and expanding the layerwise updates into polynomial terms:

$$\boldsymbol{Z} = \left( \prod_{\ell=1}^{L} ((\tilde{\theta}_0 + 1)\boldsymbol{I} + \sum_{r=1}^{|\mathcal{R}|} \tilde{\theta}_r^{(\ell)} \tilde{\boldsymbol{A}}_r) \right) \boldsymbol{X}\boldsymbol{W},$$

where $(\tilde{\theta}_0 + 1)$ corresponds to the residual connection.

This enables efficient training on large-scale HINs such as OGBN-MAG, and also permits the use of labels as attributes, as proposed by Yang et al. (2023).

## E  MUTATION PROBABILITY ANALYSIS FOR GA-BASED SCHEMA SEARCH

We conduct a parameter study on the genetic algorithm used in the systematic search for schema refinement. Figure 4 presents the performance curves over search trials under different mutation probabilities.

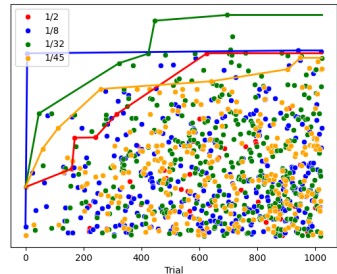

Figure 4: Performance improvement under different mutation probabilities across schema-search trials on Freebase.

Table 6: Performance comparison of sRGCN on schemas searched with different mutation probabilities.

| Mutation-prob. | Macro-F1 | Micro-F1 |
|---|---|---|
| 1/2 | $55.32{\pm}1.41$ | $66.72{\pm}0.25$ |
| 1/8 | $55.40{\pm}0.66$ | $67.32{\pm}0.66$ |
| 1/32 | $55.29{\pm}0.87$ | $67.49{\pm}0.37$ |
| 1/45 | $52.54{\pm}1.67$ | $66.34{\pm}0.46$ |

When the mutation probability is too small (e.g., $1/45$, effectively flipping only one parameter per trial), convergence becomes slower and may lead to suboptimal results. For other reasonable settings (e.g., $1/2$, $1/8$, $1/32$), the GA yields similar convergence behavior and comparable final performance. The quantitative results are summarized in Table 6.

## F    CONNECTION TO DATABASE NORMALIZATION

Attribute atomization for categorical attributes is closely related to database normalization. Database normalization decomposes tables into smaller, separate tables that satisfy specific normal forms, often to reduce redundancy and limit undesirable dependencies.

In particular, the sixth normal form (6NF) decomposes a table so that it contains a primary key and at most one additional attribute (Harrington, 2016). If a relational database is already in 6NF and all categorical attributes are represented by foreign keys pointing to reference tables that enumerate their possible values, then—under the transformation described by Wang et al. (2024)—the resulting structure corresponds directly to an atomic HIN. From this viewpoint, atomic HINs can be viewed as the graph-structured analogue of a fully decomposed schema.

When numerical attributes are present, our atomization extends this correspondence by representing numeric dimensions as attribute nodes connected via weighted edges, ensuring that all attribute information—categorical or numeric—is encoded structurally. This enables HGNNs to operate over a fully structural representation while retaining the expressive benefits of full decomposition.

## G    LIMITATIONS AND FUTURE DIRECTIONS

While our study highlights the benefits of *atomic HINs*, several limitations remain. Our treatment of attributes primarily relies on metadata, whereas, in principle, they could be divided into finer-grained entities. For example, numerical columns are currently grouped into a single node type, even though each column may carry a distinct semantic meaning. Such granularity introduces additional complexity, which remains challenging even for mainstream HGNNs.

Looking ahead, we outline several directions where *atomic HINs* may enable further advances:

**Soft relaxation of selection.**    Schema refinement in Section 4.2 is framed as binary selections $\alpha_r, \beta_\tau \in \{0, 1\}$. These can be relaxed beyond binary: node-type selectors $\beta_\tau$ could interpolate between no identity embedding ($\beta_\tau = 0$) and full-rank embeddings ($\beta_\tau = 1$), with low-rank em-

beddings corresponding to intermediate values. Similarly, edge-type selectors $\alpha_r$ could be relaxed into continuous meta-weights over relations, particularly suitable for models that parameterize heterogeneity through relation scalars.

**Inductive bias of human-defined schemas.** Heuristically constructed schemas in common benchmarks often follow practical principles: sparser node types (with larger cardinality) are modeled as structural entities without identity embeddings, while denser node types (with smaller cardinality) are left as attributes. These design choices implicitly define the inductive bias of current HGNNs and may hinder fair evaluation of new models. At the same time, counterexamples highlight the limitations of these heuristics. For instance, the *venue* relation in DBLP and the *subject* relation in ACM provide consistent gains despite their low cardinality, while high-cardinality nodes such as *keyword* or *director* in IMDb also contribute positively when modeled with ID embeddings. These cases underscore the importance of minimizing heuristic assumptions in schema design. More broadly, less conventional structures—such as bottleneck nodes or numerical attributes atomized into dense relation nodes—call for tailored architectural mechanisms in future HGNNs, rather than relying solely on adaptations of existing designs.

LARGE LANGUAGE MODELS

Large Language Models (LLMs) were used for linguistic polishing and editing of the manuscript. All technical content, theoretical results, and experiments were designed and implemented independently by the authors.

