# OpenReview forum: "Atomic HINs: Entity-Attribute Duality for Heterogeneous Graph Modeling"
_ICLR.cc/2026/Conference — ICLR 2026 Poster_

### Official Review · Reviewer_uBa4 · 2025-10-21

**Soundness:** 2
**Presentation:** 3
**Contribution:** 2
**Rating:** 4
**Confidence:** 4

**Summary:**

This paper introduces the principle of entity-attribute duality, motivating an 'atomic HIN' where all attributes become entities to maximize the graph's expressiveness. On this foundation, the authors propose a framework for task-specific schema refinement, which systematically selects the most informative nodes and relations for a given task. This allows a simple model to achieve state-of-the-art performance and stronger models to improve further, demonstrating that schema design is a critical and overlooked component in heterogeneous graph modelling.

**Strengths:**

1. The papers validates its framework across eight heterogeneous benchmark datasets from diverse domains (bibliometrics, e-commerce, knowledge graphs, biomedicine).
2. By canonicalizing schema design and turning it into a first-class, searchable object, the work shifts attention from only designing graph neural layers to co-designing data schemas + models.
3. The finding that benchmark schemas are often far from optimal (under equal model capacity) is important for the community’s evaluation practices; the release of atomic graphs, refined schemas, and a refinement framework directly supports more principled future benchmarking.

**Weaknesses:**

1. The paper formalizes attribute atomization and 'atomic HINs,' but it also acknowledges that constructing structure from attributes is a long-standing preprocessing technique (e.g., one-hot/multi-hot attributes turned into nodes/edges, as in IMDB). This makes the core idea feel more like a principled unification than a new representational primitive. It is recommended to add a small related-work ablation to show where each piece of the proposed appproach adds value beyond prior metapath/feature-node practices.
2. Schema refinement is cast as binary search with a genetic algorithm and a sizeable budget (1024 schema trials), followed by 256 model-HP trials. This raises fairness and overfitting concerns if competitor baselines do not receive commensurate tuning.
3. Several methods effectively learn metapaths/subgraphs via soft weights (e.g., GTN, MHGCN, RE-GNN). Since these are differentiable schema search to a degree, they’re natural baselines/foils for the discrete approach. If excluded due to scale or incompatibility, justify; otherwise is would be better to add them.

**Questions:**

1. What specific components constitute the core novelty beyond formalizing attribute atomization (e.g., canonical atomic HIN, schema refinement/search, pre-propagation), and how does each component independently contribute to performance?
2. What are the per-dataset costs of atomization in terms of graph expansion (nodes/edges/types), density, training time, and peak memory?
3. What are the time and memory complexities of pre-propagation as a function of relation/type counts, and how do those complexities manifest on OGBN-MAG-scale graphs?
4. How does the proposed discrete search compare empirically to differentiable schema/subgraph selection methods (e.g., GTN/MHGCN/RE-GNN) on small/medium benchmarks?
5. How transferable are refined schemas across heterogeneous GNN architectures and tasks, and what failure cases or negative transfers were observed?

---

> ### Author Response · Authors · 2025-11-21
>
> ### [W1] Related-Work suggestion
> We thank the reviewer for the insightful comment. While attribute-based structure has appeared in earlier pipelines, atomic HINs go beyond these practices in two key ways. First, atomic HINs systematically apply attribute atomization to all attributes, not only selected binary features, and do so under a theoretical principle—entity–attribute duality—that ensures symmetry and maximal expressiveness. This yields a fully explicit, canonical representation of heterogeneous data, which is not present in prior feature-node or metapath constructions.
>
> Second, our contribution extends beyond unification. The atomic form enables us to convert the otherwise open-ended problem of schema design into a well-defined, optimizable search space. Through schema refinement, we represent schema design as simple node- and edge-type selections, making it measurable, searchable, and comparable across tasks—something that prior heuristics or metapath practices do not provide.
>
> We have updated the related-work section to clarify these distinctions. As discussed further in [W3], metapath-based methods can actually benefit from improved schemas produced by our framework, underscoring the complementarity rather than redundancy of our approach.
>
>
>
> ### [W2] Fairness
>
> We appreciate the reviewer’s concern. For all baselines, we use the officially reported hyperparameters and results whenever available. When official settings are missing or evaluation protocols differ, we perform a carefully matched 256-trial hyperparameter search to ensure fairness. For example, for PSHGCN—the strongest recent baseline—we tuned hyperparameters on IMDB and Freebase using this same 256-trial procedure, while using official configurations for ACM and DBLP.
>
> Crucially, we find that schema refinement provides substantially larger gains than additional hyperparameter tuning. To verify this, we ran an extended hyperparameter search on the vanilla schema with a much larger budget (256 + 1024 trials). This led to no improvement on IMDB, DBLP, or ACM for PSHGCN, and only a small gain on Freebase (Micro-F1: 62.70 ± 0.77 → 63.25 ± 0.73). In contrast, pairing PSHGCN with our refined atomic schema yields a significantly larger improvement (62.70 ± 0.77 → 66.36 ± 0.36), indicating that the benefits stem from schema design rather than HP tuning.
>
> Although we report results using a 1024-trial schema search for completeness, we empirically observe that approximately 300 trials already achieve near-optimal performance on most datasets. Moreover, schema search is performed once per dataset, and the resulting refined schema transfers well across HGNN architectures (Section 5.5). Because the schema acts as a meta-template of the domain, not of individual data instances, it remains applicable as new nodes or edges are added.
>
>
>
> ### [W3][Q4] Differentiable schema/subgraph selection methods (GTN, MHGCN, RE-GCN)
>
> We thank the reviewer for the helpful suggestion. Differentiable methods, such as GTN, MHGCN, and RE-GCN, do perform soft metapath or subgraph selection; however, they operate over a fixed underlying schema, learning weighted compositions of existing relations rather than deciding which node types or relation types should exist in the first place. In this sense, these approaches complement our framework, which focuses on schema-level structural choices that define the space on which such methods operate.
>
> Following the reviewer’s recommendation, we have added GTN and RE-GCN to Table 1 in the revised manuscript, and we include the corresponding results below. Across datasets, we observe that applying HGNNs to refined atomic HINs yields substantially larger gains than relying solely on differentiable metapath/subgraph selection. This empirical evidence aligns with the conceptual distinction above: differentiable methods improve within a fixed schema, whereas our approach improves the schema itself, providing benefits that soft metapath weighting cannot capture.
>
> | Dataset / Metric     | GTN                | RE-GCN              | sRGCN(refined atomic)     |
> |----------------------|--------------------|---------------------|----------------------|
> | IMDB MacroF1         | 66.35 ± 0.26       | 65.94 ± 0.43       | 68.97 ± 0.09        |
> | IMDB MicroF1         | 68.93 ± 0.34       | 67.89 ± 0.43        | 71.20 ± 0.17         |
> | Freebase MacroF1     | 46.60 ± 2.25       | 40.45 ± 1.40        | 55.40 ± 1.25         |
> | Freebase MicroF1     | 63.72 ± 1.01       | 64.16 ± 0.65        | 67.32 ± 0.66         |
> | DBLP MacroF1         | 93.97 ± 0.18       | 94.80 ± 0.23        | 95.55 ± 0.19         |
> | DBLP MicroF1         | 94.43 ± 0.17       | 95.18 ± 0.21        | 95.85 ± 0.12         |
> | ACM MacroF1          | 93.32 ± 0.17       | 93.99 ± 0.41        | 94.36 ± 0.22         |
> | ACM MicroF1          | 93.24 ± 0.18       | 93.90 ± 0.42        | 94.29 ± 0.22         |

---

> ### Author Response · Authors · 2025-11-21
>
> ### [Q1] Independent Contributions of Core Components
>
> We thank the reviewer for the question. Our additional contributions beyond formalizing attribute atomization are summarized in the response to [W1]. Here, we clarify the role of each component and describe how we isolate or remove it to assess its individual contributions. Because the components are interdependent, we evaluate controlled variants that approximate the effect of disabling or enabling each one:
>
> 1. **w/o atomization**:
> Schema refinement is naturally defined on atomic HINs; applying it directly to non-atomic graphs is not well-posed because attributes remain embedded in feature vectors. To approximate a “w/o atomization” setting, we apply schema refinement directly on the vanilla HINs, which retain their original node features and do not atomize attributes into entities. Because refinement requires selecting over node types, we perform this as follows: for link prediction, refinement is applied over all node types, since no single type is privileged; for node classification, refinement is applied only over non-target node types, since the target nodes’ raw features typically carry essential information and removing them would strongly distort the task. This setup provides the closest feasible approximation of refinement without performing attribute atomization.
>
> 2. **w/o refinement:**
> We also evaluate the atomic HIN without any schema search. While full atomization alone helps on some datasets (Amazon, PubMed), it performs noticeably worse on IMDB, Freebase, and LastFM. Additionally, raw atomization introduces higher memory usage when all types are retained (e.g., DBLP), underscoring the importance of refinement for both performance and practical efficiency.
> 3. **Effect of pre-propagation:**
> Pre-propagation is designed to remove dependency issues between node- and edge-type selections during refinement. Without it, refinement can fail entirely—for example, on LastFM, the model yields empty features when refined without pre-propagation. Because disabling pre-propagation breaks refinement, we instead apply it to vanilla schemas (which are already attribute-free on Freebase and LastFM) to assess its standalone effect. As expected, its impact on performance is minimal; its primary role is to ensure the correctness and stability of refinement, rather than directly improving performance.
> We will incorporate a concise summary of these findings in the revision to clarify how each component contributes to the overall framework.
>
>
>
> | Dataset / Metric     | Vanilla            | w/o atomization      | w/o refinement      | Refined atomic      |
> |----------------------|:--------------------:|:----------------------:|:----------------------:|:----------------------:|
> | IMDB MacroF1         | 67.64 ± 0.41       | 67.96 ± 0.26         | 68.22 ± 0.39         | 68.97 ± 0.09         |
> | IMDB MicroF1         | 70.05 ± 0.50       | 70.41 ± 0.51         | 70.37 ± 0.50         | 71.20 ± 0.17         |
> | DBLP MacroF1         | 95.31 ± 0.29       | 95.26 ± 0.20         | 95.24 ± 0.14         | 95.55 ± 0.13         |
> | DBLP MicroF1         | 95.65 ± 0.30       | 95.62 ± 0.22         | 95.61 ± 0.13         | 95.85 ± 0.12         |
> | Amazon ROCAUC        | 95.94 ± 0.28       | 97.68 ± 0.08         | 97.80 ± 0.06         | 97.85 ± 0.07         |
> | Amazon MRR           | 98.43 ± 0.18       | 99.14 ± 0.05         | 99.22 ± 0.05         | 99.26 ± 0.05         |
> | PubMed ROCAUC        | 87.42 ± 0.28       | 88.76 ± 0.24         | 89.21 ± 0.88         | 90.11 ± 0.19         |
> | PubMed MRR           | 94.50 ± 0.34       | 96.12 ± 0.11         | 95.65 ± 0.77         | 96.14 ± 0.04         |
>
> | Dataset / Metric   | Vanilla          | +pre-propagation   | Refined atomic             |
> |--------------------|:------------------|:-------------------:|:-------------------:|
> | Freebase MacroF1   | 52.13 ± 1.78     | 54.10 ± 1.52      | 55.40 ± 0.66      |
> | Freebase MicroF1   | 67.09 ± 0.43     | 66.10 ± 0.72      | 67.32 ± 0.66      |
> | LastFM ROCAUC      | 70.55 ± 0.15     | 70.93 ± 0.17      | 77.10 ± 0.17      |
> | LastFM MRR         | 91.25 ± 0.57     | 91.56 ± 0.27      | 93.70 ± 0.16      |

---

> ### Author Response · Authors · 2025-11-21
>
> ### [Q2] Atomization cost and its effect on graph size, density, training time, and memory
>
> We thank the reviewer for the question. The atomization cost is negligible and incurred only once per dataset. Atomization simply converts each attribute into edges (one per entity–attribute association), so the cost depends only on the increase in edge count relative to the original graph. The statistics comparing vanilla and atomic schemas for each dataset are provided in Table 4. For reference, performing atomization on the large-scale OGBN-MAG takes 39 seconds.
>
> The effects on density, training time, and peak memory are summarized in the table below. Because these metrics depend on the specific schema and HGNN used, we report the comparison between the vanilla schema and the final refined schema under sRGCN. As shown, the changes in density and memory are modest, and training times remain comparable to standard HGNN pipelines.
>
> | Dataset | Mem (vanilla) | Mem (refined) | Time (vanilla) | Time (refined) | Density (vanilla) | Density (atomic) |
> |---|-------:|------:|-------:|-------:|---------:|--------:|
> | IMDB     |   3.2 |   8.0 |   0.4 |   0.7 | 1.90E-04 | 5.59E-04 |
> | Freebase |  20.8 |  17.3 |   1.5 |   1.2 | 6.52E-05 | 6.52E-05 |
> | DBLP     |   5.6 |   8.9 |   0.4 |   0.6 | 3.51E-04 | 1.37E-03 |
> | ACM      |   5.6 |   5.6 |   0.4 |   0.4 | 4.66E-03 | 4.66E-03 |
> | MAG      |  23.6 |  24.2 |  13.5 |  14.1 | 1.10E-05 | 6.20E-05 |
> | Amazon   |   1.7 |   2.1 |   0.8 |   0.9 | 2.38E-03 | 2.70E-03 |
> | LastFM   |   1.6 |   1.0 |   0.7 |   0.6 | 5.26E-04 | 5.26E-04 |
> | PubMed   |   2.9 |   4.3 |   0.1 |   0.1 | 1.17E-04 | 1.44E-03 |
>
> ### [Q3] Complexities of pre-propagation
>
> We thank the reviewer for the question. Pre-propagation is a one-time preprocessing step and adds no runtime overhead during model training. Its cost comes from propagating one-hot identity signals across all relations. The complexity of pre-propagation is loosely bounded by $O(|\mathcal E||\mathcal T|)$, where $|\mathcal E|$ is the number of edges and $|\mathcal T|$ is the number of node types. The $|\mathcal E|$ term corresponds to a single one-hot propagation over the full graph, and $|\mathcal T|$ reflects the maximum possible propagation distance.
>
> In practice, the cost is modest. On the atomic OGBN-MAG graph, pre-propagation completes in 188 seconds on the CPU, and this process is performed only once for the entire dataset.
>
> We have added a dedicated subsection (Section 5.7) in the revision that breaks down the complexity of each component, including how this bound arises for pre-propagation.
>
>
>
> ### [Q5] Transferability of refined schemas
> We thank the reviewer for the question. As shown in Section 5.5 and Table 3, we assess transferability by applying schemas refined with sRGCN to other HGNN architectures. Across all datasets, we observe consistently positive transfer: SimpleHGN and PSHGCN both benefit from schemas refined by sRGCN, and the magnitude of improvement is often comparable to (and sometimes larger than) their own refinements.
>
> We also find that schemas refined by different HGNNs tend to be highly similar to one another and differ much more from their respective vanilla schemas. This suggests that the refinement process captures a stable, task-relevant structure that generalizes across heterogeneous GNN architectures. We will add a brief note in the revision clarifying these observations.
>
> Regarding transfer across tasks, schema refinement is designed as a task-specific procedure. As such, we expect that schemas optimized for one task will not necessarily be optimal for a substantially different task. This is not a limitation of the method but a natural property of task-specific structural design.

---

### Official Review · Reviewer_eaMx · 2025-10-25

**Soundness:** 3
**Presentation:** 3
**Contribution:** 3
**Rating:** 6
**Confidence:** 2

**Summary:**

The paper proposes an atomic view of heterogeneous information networks (HINs) by atomizing attributes into explicit nodes and relations, so that all information becomes structural. On top of this representation, the authors introduce schema refinement via a genetic algorithm that searches over which attribute-derived node/edge types to include for a downstream task.

**Strengths:**

- Elevating attributes to first-class graph elements clarifies a long-standing, often ad-hoc HIN design step. The discussion is genuinely interesting and important for the community.
- Using GA to navigate the combinatorial schema space is a neat and pragmatic idea.
- Framing schema choice as an optimization target (rather than a one-off manual decision) is valuable and underexplored.
- Proofs and experiments are commendable

**Weaknesses:**

- Baseline: How does the proposed method compare with the tabular learning baseline?
- Additional ablation study and details over the GA could be added, i.e., different variants of the fitness function
- Scalability analysis is missing: what is the atomization cost, searching cost, and runtime/memory of the model added up, and how do it compare with standard GNN methods?

**Questions:**

- What is the connection between atomic representation and normalization form in database theory?
- Does this method extend to relational databases? Could you apply the same idea to solve the relational deep learning problem like in RelBench?

---

> ### Author Response · Authors · 2025-11-21
>
> ### [W1] Tabular learning baselines
>
> We thank the reviewer for raising this point. Tabular learning is indeed a competitive baseline on certain datasets where nodes have rich and informative features. Nevertheless, HGNN can further exploit the relational and multi-type structure in heterogeneous graphs, which are critical for HIN tasks. This difference also motivates our approach: atomization exposes all potential relations in HINs to HGNNs, while schema refinements identify the subset most relevant to the task. As a result, HGNNs equipped with atomic HINs typically outperform tabular learning approaches on standard HIN benchmarks. Following this suggestion, we have added XGBoost, a representative tabular learning baseline, to Table 1. As shown below, XGBoost is relatively competitive on IMDB, where target nodes contain richer attributes, but it is substantially weaker on datasets where structural and relational information play a more central role, such as DBLP and ACM.
>
>
> | Dataset / Metric      | XGBoost             | sRGCN (refined on atomic) |
> |-----------------------|---------------------|----------------------------|
> | IMDB MacroF1          | 64.27 ± 0.21        | 68.97 ± 0.09               |
> | IMDB MicroF1          | 67.58 ± 0.20        | 71.20 ± 0.17               |
> | Freebase MacroF1      | N/A                 | 55.40 ± 1.25               |
> | Freebase MicroF1      | N/A                 | 67.32 ± 0.66               |
> | DBLP MacroF1          | 77.96 ± 0.37        | 95.55 ± 0.19               |
> | DBLP MicroF1          | 78.65 ± 0.34        | 95.85 ± 0.12               |
> | ACM MacroF1           | 86.92 ± 0.32        | 94.36 ± 0.22               |
> | ACM MicroF1           | 86.90 ± 0.30        | 94.29 ± 0.22               |
>
>
> ### [W2] Additional ablation and GA details
>
> We thank the reviewer for the suggestion. In our setting, the goal of schema refinement is to identify schemas that improve downstream performance, so we follow the standard practice of using validation performance as the fitness signal. Exploring alternative objectives is certainly possible, but it would lead to a different problem formulation rather than an ablation of our current approach.
>
> Regarding additional GA details, we followed this suggestion and investigated the effect of mutation probabilities on the algorithm. For datasets with many schema parameters, we set the mutation probability to $1/8$; for those with fewer parameters, we use $1/(|\mathcal{R}| + |\mathcal{T}|)$. We analyzed this on Freebase, the dataset with the most complex schema.
> We find that mutation rates that are too small (e.g., flipping only one bit per trial) slow down the convergence and may yield suboptimal results. For other reasonable choices (e.g., $1/2$, $1/8$, $1/32$), the GA shows stable convergence and similar final performance, indicating that the search procedure is robust to mutation settings.
>
> The numerical results are presented in the table below, and a performance comparison across different mutation probabilities during the search trials is provided at the following link: https://anonymous.4open.science/r/AtomHIN-Rebuttal-1B35/assets/ga-mutation.png.
>
>
> | Mutation Prob. | Macro-F1           | Micro-F1           |
> |----------------|--------------------|--------------------|
> | 1/2            | 55.32 ± 1.41       | 66.72 ± 0.25       |
> | 1/8            | 55.40 ± 0.66       | 67.32 ± 0.66       |
> | 1/32           | 55.29 ± 0.87       | 67.49 ± 0.37       |
> | 1/45           | 52.54 ± 1.67       | 66.34 ± 0.46       |
>
>
> ### [W3] Scalability analysis
> We thank the reviewer for raising this point. In response, we have added a dedicated section for complexity analysis (Section 5.7). The atomization step adds negligible overhead and is performed once per dataset. Pre-propagation is also computed offline. The subsequent type-selection stage only indexes into precomputed structures, so its cost is minimal. As a result, runtime and memory are governed primarily by the refined schema and the chosen HGNN, rather than by the atomization process.
>
> Section 5.7 provides concrete illustrations of how node- and edge-type selections affect model complexity. For example, the sRGCN used in Table 1 has complexity $O(MD)$, where $M$ is the number of selected relation types and $D$ is the number of nodes in the selected types.
>
> Regarding search cost, it scales linearly with the trial budget: each trial trains a single HGNN under a candidate schema, resulting in an overall cost of $O(B)$ for $B$ trials. As shown in Section 5.6, the GA converges quickly in practice, and the total cost remains affordable.

---

> ### Author Response · Authors · 2025-11-21
>
> ### [Q1] Connection to normalization theory
>
> We thank the reviewer for the insightful question. The atomic representation is conceptually related to ideas from database normalization, particularly sixth normal form (6NF), where each table is fully decomposed so that a relation contains a primary key and at most one additional attribute [1].
>
> If a relational database is already in 6NF and contains only categorical attributes, then following the transformation described by Fey et al. [2]—mapping each row to a node and each primary–foreign-key pair to an edge—produces a structure that corresponds directly to an atomic HIN. In this sense, atomic HINs can be viewed as the graph-structured analogue of a fully decomposed schema.
>
> When numerical attributes are present, our atomization extends this mapping by representing numeric dimensions as attribute nodes connected via weighted edges, ensuring that all attribute information—categorical or numeric—is encoded structurally. This allows heterogeneous GNNs to operate over a unified structural representation while retaining the expressive benefits of full decomposition.
>
> ### [Q2] Applicability to relational databases
>
> It has been demonstrated that relational databases can be systematically converted into HINs [2], making our framework directly applicable. In this setting, atomic HINs and schema refinement offer a principled approach to reasoning about task-specific schema design on top of relational data, rather than relying on fixed database schemas.
>
> We therefore believe that the proposed approach offers a useful foundation for future HGNNs operating on large-scale relational systems—including settings such as RelBench—by enabling models to exploit structural choices that were previously treated as fixed.
>
>
> [1] Harrington, J. L. (2016). Normalization (Chap. 7). In J. L. Harrington (Ed.), Relational database design and implementation (4th ed., pp. 141–161). Morgan Kaufmann.
>
> [2] Fey, M., Hu, W., Huang, K., Lenssen, J. E., Ranjan, R., Robinson, J., Ying, R., You, J., & Leskovec, J. (2024). Position: Relational deep learning – Graph representation learning on relational databases. In Proceedings of the Forty-first International Conference on Machine Learning.

---

> ### Comment · Reviewer_eaMx · 2025-11-21
>
> Thank you for the detailed response. My concern has been fully addressed. I will raise my score from 6 to 8 for this paper. Please consider including the additional experiments and insight for the ablation study, and the connection to database theory in the updated manuscripts (Appendix is ok).  Nice work!

---

> > ### Author Response · Authors · 2025-11-21
> >
> > Thank you very much for the thoughtful follow-up and for raising your score. We sincerely appreciate the positive assessment and your constructive suggestions. We will incorporate the additional experiments, ablation insights, and the connection to database theory into the updated manuscript (with details placed in the Appendix as recommended).

---

### Official Review · Reviewer_8dUP · 2025-10-26

**Soundness:** 3
**Presentation:** 3
**Contribution:** 3
**Rating:** 4
**Confidence:** 3

**Summary:**

The work explores the new structure of HIN, which transforms certain feature values into new node types.

To this end, the authors propose an atomic HIN, which selectively transforms certain features into nodes via an evolutionary algorithm.

The authors demonstrate the effectiveness of the atomic HIN in various heterogeneous graph benchmark datasets.

**Strengths:**

S1. The motivation and presentation are clear and intriguing.

S2. Key arguments are supported by theoretical analysis.

S3. Various datasets have been used for evaluations.

**Weaknesses:**

I have several questions regarding this work:

**W1. [Complexity]** While the authors use an evolutionary algorithm to avoid the exhaustive exponential search regarding which features to be transformed into nodes, I still think the search process requires heavy computations. Can the authors analyze the time consumption for this search process?

**W2. [Backbone HIN]** It seems the proposed method is coupled with SHGC. Can the method be coupled with other types of HINs?

**W3. [Feature constraint]** I think for binary features, the transformation is natural. However, for numeric features, how is the transform being performed? Are the values assigned by edge types (or weights)? If so, then wouldn't the graph be very dense?

My initial score is below the acceptance threshold, but I’m willing to raise it pending the authors’ clarifications.

**Questions:**

See Weakness.

---

> ### Author Response · Authors · 2025-11-21
>
> ### [W1] Complexity
>
> We thank the reviewer for the question. The search complexity is $O(B)$, where $B$ is the number of trials in the genetic search. In each trial, our algorithm first updates the schema and then trains a lightweight sRGCN model accordingly, with the latter part being the major cost in a trial.
>
> In practice, the cost is affordable. On a single RTX 3060 GPU, per-trial training times are 155 seconds (IMDB), 28 seconds (ACM), and 97 seconds (DBLP). Trials can be parallelized across multiple GPUs to reduce wall-clock time.
> The search converges quickly in most cases. As shown in Section 5.6, we already achieve near-optimal performance on most datasets in less than 300 trials, which is well below the exponential search space.
> Finally, schema search can be done offline and only needs to be performed once per dataset. As demonstrated in Section 5.5, the resulting refined schema transfers effectively across HGNN architectures, and it remains valid even as new nodes or edges are added, since the schema captures the domain structure rather than instance-level data.
>
>
> ### [W2] Backbone HIN
> We would like to emphasize that the proposed framework is compatible with a wide range of HGNN architectures and diverse forms of heterogeneous information networks (HINs). Specifically, as shown in Section 4.5 and Appendix C, many representative HGNNs—both classic and recent—can be viewed as special cases or first-order approximations of SHGC. More importantly, Section 5.5 and Table 3 demonstrate that schemas refined with sRGCN transfer directly to other HGNNs such as SimpleHGN and PSHGCN, and these backbones obtain consistent additional gains. This confirms that the framework is backbone-agnostic and does not rely on any specific encoder.
>
> Additionally, the proposed framework can be applied to a wide range of HIN datasets and corresponding tasks. Specifically, our evaluation covers diverse heterogeneous networks from bibliometrics, e-commerce, knowledge graphs, social networks, and biomedicine, spanning both node classification and link prediction tasks. Across all eight datasets, the framework yields consistent improvements, indicating strong generalization across different types of HINs.
>
>
>
> ### [W3] Feature constraint (numeric features)
>
> We thank the reviewer for the question. Numeric features are treated analogously to binary features: each feature dimension is atomized into a node (e.g., a 128-dimensional attribute yields 128 atomized nodes), and the original numeric value is represented as an edge weight. This yields a star-like structure centered at each attribute-node: every original node has one weighted edge to each relevant attribute-node. While this forms a locally dense subgraph around the attribute-nodes, it does not make the entire graph dense.
>
> Somewhat surprisingly, schema refinement often preserves these attribute-induced relations in datasets with rich numeric attributes (e.g., IMDB, OGBN-MAG, Amazon), suggesting that they encode useful signals (Obs. 3, Section 5.3).
> We have revised the manuscript to clarify how numeric attributes are atomized and why the resulting structures can be beneficial.

---

> ### Comment · Reviewer_8dUP · 2025-11-21
>
> Dear Authors,
>
> Thank you for the responses.
> My main concerns have been addressed, and therefore, I have raised my score from 4 to 6.
>
> Please incorporate these clarifications to the manuscript.

---

> > ### Author Response · Authors · 2025-11-21
> >
> > Thank you very much for your thoughtful follow-up and for raising your score. We truly appreciate your time, feedback, and constructive suggestions. We will incorporate all clarifications into the revised manuscript as recommended.

---

### Official Review · Reviewer_c825 · 2025-10-29

**Soundness:** 4
**Presentation:** 3
**Contribution:** 2
**Rating:** 4
**Confidence:** 3

**Summary:**

This paper works on heterogeneous information networks for which many different machine learning techniques are known. These models start with the data and their structure as given. The authors argue that there is in fact a lot of different ways the data can be structured into HINs and this can have a high impact on the results. They define a formal framework to show the entity - attribute duality which leads to the possibility to do this. In two propositions the show that two existing methods (from 2018 and 2019) can be expressed within this framework. The results indicate that indeed the results vary with the choice of entities vs attributes and that the revised schemas proposed by the authors have good performance.

**Strengths:**

- Good theoretical foundation for the entity - attribute duality and the following atomization step.
- Unification of existing methods based on the framework.
- Good experimental evaluation

**Weaknesses:**

- This paper is more a database / KDD type of paper, for me that would be the more appropriate venues. The authors are correct by writing that ICLR typically start from the given structure. This is also reflected in the references where the only refs to ICLR and related venues are for the tools used not as comparisons to the proposed methods.
- The reformulation of two methods into the new setting is interesting but the methods are from 2018 and 2019

**Questions:**

- How about more recent methods compared to the 2018 and 2019 ones. Why only consider those two?

**Details Of Ethics Concerns:**

N.A.

---

> ### Author Response · Authors · 2025-11-21
>
> ### [W1] Clarification on Database/Dataset Positioning
>
> We thank the reviewer for the comment. We respectfully clarify that our contribution is not dataset construction, but a theoretical and model-level framework for heterogeneous graph learning. Our main contribution lies in introducing atomic HINs as a canonical and theoretically grounded representation, together with schema refinement, which turns schema design into a principled structural learning problem. These contributions directly influence the expressive power and inductive bias of HGNNs, and we validate them empirically across eight benchmarks. The release of the atomic HINs dataset and refined schemas is intended to support transparency and reproducibility, rather than serving as a primary contribution.
>
>
> Regarding the comment that *“ICLR typically starts from the given structure,”* our work specifically addresses this long-standing assumption by showing that the schema itself is a critical modeling dimension that significantly impacts HGNN performance.
> Regarding the statement that *“the only refs to ICLR and related venues are for the tools used, not as comparisons to the proposed methods”*, we would appreciate clarification. The meaning of this observation is not entirely clear to us. We compare our results against widely used and representative HGNN baselines from multiple venues, following the standard practice in heterogeneous graph learning. Understanding more precisely which comparisons the reviewer has in mind would help us address this point appropriately in the revision.
>
>
>
> ### [W2, Q1] Scope of Reformulated Methods (2018–2019)
>
> We thank the reviewer for the comment. We would like to emphasize that the reformulation is very general and can support a variety of HGNNs. Besides the two methods reformulated in the main text, we also include the reformulation of recent representative HGNNs—SimpleHGN (2021), SeHGNN (2023), and PSHGCN (2024)—in Appendix C and Table 5. These reformulation examples, covering a broad range of HGNN architectures, demonstrate the flexibility and generalization of the proposed reformulation. We chose RGCN and GTN as main-text examples because they remain among the most widely used and influential HGNNs (7,345 and 1,677 citations), and they represent the core modeling paradigms on which many later methods build.  We have revised the main text to better reflect this point.

---

> > ### Comment · Reviewer_c825 · 2025-11-27
> >
> > The intention was not to say that this was dataset construction. In my opinion this work is more database schema construction where you show that the chosen schema has a clear impact on the result. As concerns the references to ICLR, what I meant there is that the ICLR references are on the models used, these are not on schema construction. I did miss though that Fey to which you compare in the introduction is ICML. Hence, I will update my score.

---

> > > ### Author Response · Authors · 2025-11-27
> > >
> > > Thank you for the clarification and for updating your score. We appreciate your thoughtful feedback.

---

### Meta-Review · Area_Chair_GEQC · 2026-01-07

**Summary:**

**Summary:**
This paper introduces the concept of ``atomic heterogenous information networks'' which seek to capture the entity attribute duality. This is done through the use of a schema refinement framework that uses a genetic algorithm based search to select optimal node and edge types for downstream tasks. They demonstrated that a relational gcn can be trained on refined atomic HINs to achieve iota performance for node classification and link prediction.

**Rationale:**
This paper makes a meaningful contribution to the underexplored but important field of heterogeneous graph learning, and specifically to the systematic design of graph schemas. While attribute atomization has appeared in prior preprocessing pipelines, the authors provide the first principled formalization through entity-attribute duality, converting the open-ended schema design problem into a well-defined, searchable optimization space. The paper is well written with clear experiments that show the benefits of this construction. The rebuttal period was unusually successful with multiple of the concerns so thoroughly addressed that multiple reviewers bumped their scores up multiple points. Having read the reviews, I feel that this was quite deserving and as such, I feel compelled to recommend acceptance.

**Reviewer Concerns:**

c825:
- positioning: the work feels more like a database/schema paper, maybe inline for kdd. Addressed
- backbone: rgcn/gtn backbones are old, are they relevant to modern architectures. Addressed
- limited comparisons to recent ICLR/NeurIPS/ICML methods. Addressed

8dUP:
- complexity: evolutionary search process is heavy weight. Addressed
- unclear how the method couples with HGNNs. Addressed
- Numeric features? Addressed

eaMx:
- tabular baselines: addressed
- scalability analysis: addressed
- Comparisons with differentiable schema baselines: addressed

uBa4:
- Novelty: Addressed, but not convincingly
- Tuning fairness:
- Missing baselines: addressed
- per-dataset costs of atomization: addressed
- questions about transferability: addressed

**Reviewer Scores:**

- c825, 4 -> 5, The reviewer was concerned about relevance to the venue but the authors clarified this point. Even still positioning concerns remain
- 8dUP, 4 -> 6, Main concerns about complexity remain but numeric features and backbone compatibility were addressed
- eaMx, 6 -> 8, The concerns were addressed and the score was bumped accordingly
- uba4, 4->5, the concerns about novelty weren't addressed convincingly but everything else was addressed well

---

### Decision · Program_Chairs · 2026-01-26

Accept (Poster)